# Sequential actions of EOMES and T-BET promote stepwise maturation of natural killer cells

Jiang Zhang [1,2], Stéphanie Le Gras[3,4], Kevin Pouxvielh[1], Fabrice Faure[5], Lucie Fallone[1], Nicolas Kern[1], Marion Moreews[1], Anne-Laure Mathieu[1], Raphaël Schneider[6], Quentin Marliac [1], Mathieu Jung[3,4], Aurore Berton[1], Simon Hayek[7], Pierre-Olivier Vidalain [1,7], Antoine Marçais[1], Garvin Dodard[8], Anne Dejean [9], Laurent Brossay [8], Yad Ghavi-Helm [6] & Thierry Walzer [1✉]

EOMES and T-BET are related T-box transcription factors that control natural killer (NK) cell development. Here we demonstrate that EOMES and T-BET regulate largely distinct gene sets during this process. EOMES is dominantly expressed in immature NK cells and drives early lineage specification by inducing hallmark receptors and functions. By contrast, T-BET is dominant in mature NK cells, where it induces responsiveness to IL-12 and represses the cell cycle, likely through transcriptional repressors. Regardless, many genes with distinct functions are co-regulated by the two transcription factors. By generating two gene-modified mice facilitating chromatin immunoprecipitation of endogenous EOMES and T-BET, we show a strong overlap in their DNA binding targets, as well as extensive epigenetic changes during NK cell differentiation. Our data thus suggest that EOMES and T-BET may distinctly govern, via differential expression and co-factors recruitment, NK cell maturation by inserting partially overlapping epigenetic regulations.

[1] CIRI, Centre International de Recherche en Infectiologie, Univ Lyon, Inserm, U1111, Université Claude Bernard Lyon 1, CNRS, UMR5308, ENS de Lyon, F-69007 Lyon, France. [2] Shanghai Key Laboratory of Regulatory Biology, School of Life Sciences, East China Normal University, Shanghai, China. [3] IGBMC, CNRS UMR7104, Inserm U1258, Université de Strasbourg, Illkirch, France. [4] Plateforme GenomEast, infrastructure France Génomique, Illkirch, France. [5] Institut NeuroMyoGène, INSERM U1217/CNRS UMR5310, Université de Lyon, Université Claude Bernard, Lyon 1, Lyon, France. [6] Institut de Génomique Fonctionnelle de Lyon, CNRS UMR 5242, Ecole Normale Supérieure de Lyon Université Claude Bernard Lyon 1, 46 allée d'Italie, F-69364 Lyon, France. [7] Equipe Chimie et Biologie, Modélisation et Immunologie pour la Thérapie (CBMIT), Université Paris Descartes, CNRS UMR 8601, 75006 Paris, France. [8] Department of Molecular Microbiology and Immunology, Division of Biology and Medicine, Brown University Alpert Medical School, Providence, RI 02912, USA. [9] Institut Toulousain des Maladies Infectieuses et Inflammatoires (Infinity), INSERM UMR1291 - CNRS UMR5051 - Université Toulouse III, Toulouse, France. ✉email: thierry.walzer@inserm.fr

Natural Killer (NK) cells are group 1 innate lymphoid cells (ILCs) with an important role in antiviral[1] and anti-tumor responses[2]. NK cells share many features with tissue-resident ILC1s, such as the responsiveness to IL-15, IL-12 and IL-18, and the capacity to rapidly produce IFN-γ upon stimulation[3]. However, they differ from ILC1s by their capacity to circulate in the blood, by their expression of multiple receptors of the Ly49 family, by their higher cytotoxic potential and by their expression of integrin subunits. In particular, CD49B and CD49A are expressed in a mutually exclusive manner by NK cells and ILC1s, respectively[4,5]. Moreover, NK cells and ILC1s are developmentally distinct. Indeed, even though all ILCs share a common progenitor, NK cells rapidly branch out from the main ILC developmental pathway[6], and the factors that promote this route remain unclear. NK cells then operate a process of maturation that starts in the bone marrow (BM) and continues in the periphery. This process includes at least three discrete stages that can be discriminated by surface expression levels of CD11B and CD27[7,8]. CD11B⁻ CD27⁺ (hereafter referred as CD11B⁻) are the most immature NK cells, and give rise to CD11B⁺ CD27⁺ (double positive, DP) which then differentiate into CD11B⁺ CD27⁻ (CD27⁻). Mature NK cells (either DP or CD27⁻) are more cytotoxic than immature ones against tumor targets[8] and express a distinct set of trafficking molecules that allow them to circulate in the blood. In particular, they express the sphingosine-1 phosphate receptor S1PR5, which promotes exit from lymphoid organs[9,10].

The specification of immune lineages depends on a network of lineage-determining TFs that induce hallmark genes and repress the expression of other lineages, thereby restricting pluripotency as cells differentiate[11]. NK cell development and maturation is orchestrated by a network of TFs including the related T-box TFs T-BET and EOMES[12,13]. EOMES and T-BET are thought to have similar DNA binding properties, owing to their highly homologous T-box DNA binding domains, which are 74% identical. Large scale chromatin accessibility analysis across immune subtypes[14–17] predicted a major role of T-box TFs in the regulation of NK-cell specific enhancers. However, they failed to discriminate between T-BET and EOMES in this role. $Tbx21$ (that encodes for T-BET) was cloned in 2000 and immediately recognized as an essential driver of Th1 differentiation and IFN-γ production in CD4 T cells[18]. The analysis of $Tbx21^{-/-}$ mice revealed an essential role for T-BET in NK cell homeostasis and function[19]. In these mice, NK cells display a higher turnover associated with higher apoptosis rate[19], and an immature phenotype C-KIT^{high} CD43^{low} [19], KLRG1⁻ [20] and CD27⁺ [21]. A more recent study suggested that T-BET stabilizes immature NK cell attributes, somewhat contradicting previous findings[22]. As recognized later, however, this conclusion was reached after using an incorrect NK cell gating strategy that also included ILC1s[4]. T-BET is indeed essential for the development of ILC1s, whose phenotype is highly similar to that of immature NK cells[4]. Moreover, a recent single cell RNA-seq analysis firmly established that T-BET suppresses the immature NK cell transcriptional signature[23]. Functionally, T-BET is important to promote NK cell responsiveness to IL-12[19] and to support their blood circulation through S1PR5[24], which could be important for their capacity to control lung metastases in the B16 model[25,26]. In addition, T-BET promotes the survival of mouse cytomegalovirus-specific memory NK cells[27].

The function of EOMES was first studied in CD8⁺ T cells, where it was shown to promote the expression of the IL-15 receptor subunit Il2Rβ (or CD122), together with T-BET[28]. Therefore, EOMES and T-BET were proposed to redundantly regulate the differentiation of CD8⁺ effector T cells. Later in vitro studies showed a strong correlation between EOMES and

PERFORIN expression in CD8⁺ T cells activated with antigen and cytokines[29,30]. This led to the concept that EOMES rather than T-BET, drives the cytotoxic phenotype. Generation of floxed $Eomes$ alleles allowed conditional deletion of $Eomes$ in immune cells, circumventing the lethality issue of $Eomes$ knockout mice. Deleting $Eomes$ in VAV1⁺ immune cells[22], in NKP46⁺ cells[31] or in NKP46⁺ cells in an inducible manner[32], compromised NK cell development. In the latter system, the authors found that EOMES was required to preserve NK cell viability, especially at the CD27⁺ CD11B⁺ stage, and that it was essential for cytotoxicity but not for IFN-γ secretion[32]. How EOMES promotes NK cell development i.e., what genes are specifically regulated by this TF remains however unclear. The nature of EOMES and T-BET cooperation is another unresolved question. Do they play redundant roles or do they have specific and complementary functions? Overexpression of EOMES rescues IFN-γ production by T-BET deficient CD4⁺ T cells[33,34]. Moreover, the combined deletion of $T-BET$ and $EOMES$ leads to a complete deficiency in NK cells while single mutations of either gene leads to a partial defect[22]. Together with the similarity in EOMES and T-BET DNA binding domains, these data argue for a redundancy between both factors. Yet, the direct and indirect EOMES targets in NK cells are mostly unknown, as only a few studies have assessed their DNA binding properties, but not in resting NK cells, and not in a comparative manner with T-BET. ChIP-seq analyses would represent a significant advance. However, such analyses remain undeniably complicated in primary cells. Thus, the cellular and molecular roles of T-BET and EOMES during NK cell development and maturation remain to be clarified.

Here we present a comprehensive analysis of the role of EOMES and T-BET during NK cell development. We show a dominant expression and a dominant role of EOMES in immature NK cells and reciprocally of T-BET in mature ones. EOMES is required to specify the NK cell lineage and promote the survival of immature NK cells while T-BET is required for terminal differentiation but is not involved in early development. Using newly generated mouse models expressing endogenously tagged T-BET and EOMES, we performed genome-wide analysis of T-BET and EOMES binding. We also uncover large epigenetic changes at genomic sites of EOMES and/or T-BET binding during NK cell differentiation, suggesting that both TFs cooperate to shape the epigenetic landscape of mature NK cells.

## Results

**EOMES and T-BET balance each other during NK cell maturation and activation.** To explore the roles of T-BET and EOMES in NK cells, we first measured the expression of both TFs during maturation. To unambiguously analyze NK cells and exclude ILC1s we defined NK cells as CD49B⁺ CD49A⁻ NK1.1⁺ CD3⁻ and analyzed maturation based on the CD11B/CD27 classification (Supplementary Fig. 1A). Flow cytometry measurements indicated that EOMES expression in spleen NK cells was rather similar in stages 1–2 and then decreased in stage 3 NK cells, while T-BET had the reciprocal expression pattern (Fig. 1A). In the BM, EOMES levels showed the same pattern as in the spleen while T-BET levels were much lower compared to that of spleen NK cells (Supplementary Fig. 1B), as we previously observed[4].

We used $Tbx21^{-/-}$ mice and $NCR1-iCre^{+/-}$ $Eomes^{fl/fl}$ mice (referred to as $NK-Eomes^{-/-}$ thereafter) to study the consequences of T-BET or EOMES deficiency on NK cell development. We noted that in mice with one copy of $Eomes$, NK cells had near normal levels of EOMES protein, while in mice with one copy of $Tbx21$, the level of this TF was approximately half of that of WT mice (Fig. 1A). In T-BET deficient NK cells (i.e., from $Tbx21^{-/-}$ mice), a higher expression of EOMES was also noted in all subsets,

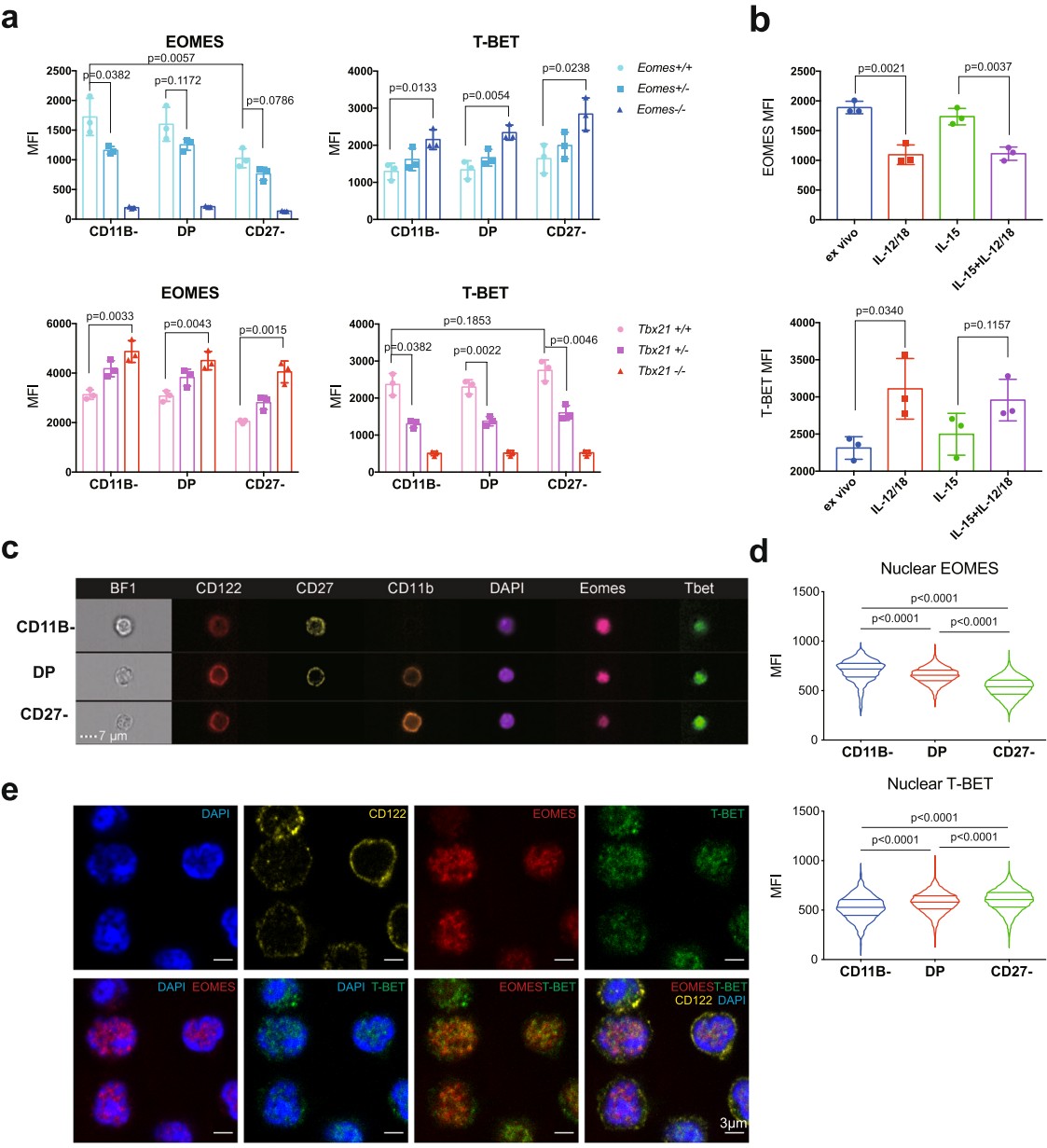

**Fig. 1 EOMES and T-BET balance each other during NK cell maturation. a** Flow cytometry measurement of T-BET and EOMES in gated NK cells from WT, *Tbx21*[+/−], *Tbx21*[−/−], *Ncr1*[Cre/+] (*Eomes*[+/+]), *Ncr1*[Cre/+] X *Eomes*[lox/+] (*Eomes*[+/−]) and *Ncr1*[Cre/+] X *Eomes*[lox/lox] (*Eomes*[−/−]) mice, as indicated. Bar graphs show the mean ± SD fluorescence intensity (MFI) of EOMES and T-BET staining in gated NK cell subsets from spleen. Data are representative of 3 experiments and 3 mice are shown for each group. **b** WT spleen cells were stimulated O/N in the indicated conditions and T-BET and EOMES expression were measured by flow cytometry. Bar graphs show the mean ± SD fluorescence intensity (MFI) of EOMES and T-BET staining in gated NK cells. Data are representative of 2 experiments and 3 mice are shown for each group. **c** Image stream X (ISX) analysis of T-BET and EOMES expression in gated NK cells. Representative images are shown for the three NK cell subsets. CD122 staining was used to visualize the membrane. **d** Quantification of nuclear T-BET and EOMES in gated NK cell subsets analyzed with IDEAS software. Bar graphs show nuclear MFI of T-BET and EOMES for the indicated subsets, normalized to the level of each TF in immature CD11B- NK cells. N = 1232, 5157 and 22403 cells analyzed for CD11B-, DP and CD27- subsets, respectively. **e** NK cells were sorted from WT mice in immature or mature subsets and subsequently stained for nucleus (DAPI), T-BET, EOMES and CD122 for confocal microscopy analysis. Shown are representative images of each staining and combinations for immature NK cells. Images are representative from three individual experiments. Unpaired *t* tests (two-tailed) were used for statistical analysis of data presented in this figure.

while EOMES-deficient NK cells (i.e., from *NK-Eomes*[−/−] mice) displayed higher T-BET expression (Fig. 1A). This suggested that EOMES and T-BET repress transcriptionally each other in NK cells or that the absence of one TF stabilizes the protein expression of the other one. Upon stimulation with cytokines ex vivo, IL-15 maintained T-BET and EOMES levels while a combination of IL-12 and IL-18 (together or not with IL-15) resulted in a strong up regulation of T-BET and a decrease of EOMES, further supporting

the concept that EOMES and T-BET balance each other during NK cell maturation or activation (Fig. 1B).

A previous article reported that T-BET and EOMES were localized both in the nucleus and in the cytoplasm of memory CD8[+] T cells[35], which adds another layer of regulation for both TFs. To analyze the nuclear expression of EOMES and T-BET, we used Image cytometry (Image-Stream) and confocal microscopy. Image-Stream analysis allowed us to quantify the nuclear fraction

of EOMES and T-BET, delimited by the DAPI staining (Fig. 1C). Using this method, we found that NK cell maturation was associated with a progressive shift in the nuclear ratio between EOMES and T-BET, from high EOMES/low T-BET to low EOMES/high T-BET (Fig. 1D). A confocal microscopy analysis then showed that EOMES and T-BET are predominantly nuclear in NK cells, and suggested that they localize in the same nuclear areas both in immature (Fig. 1E) and mature NK cells (Supplementary Fig. 1C, see also correlative analysis in Supplementary Fig. 1D).

**EOMES is essential for early NK cell development and restrains T-BET-induced terminal maturation.** A previous analysis of the role of T-BET and EOMES in NK cell development was hampered by a gating analysis that did not discriminate between NK cells and ILC1[22]. We therefore revisited the role of both TFs in NK cell development and maturation using $Tbx21^{-/-}$, $NK\text{-}Eomes^{-/-}$ and appropriate control mice by carefully gating NK cells (CD49B$^+$ CD49A$^-$) and excluding ILC1 (CD49A$^+$CD49B$^-$) from the analysis (see Supplementary Fig. 1). Both T-BET and EOMES were important for NK cell homeostasis (Fig. 2A). However, EOMES had a more important role than T-BET in terms of NK cell numbers, i.e., the lack of EOMES caused the number of NK cells to drop by 5–10 fold both in BM and spleen, while the lack of T-BET rather increased the number of BM NK cells, and decreased that of spleen NK cells two to threefold (Fig. 2B). When examining maturation stages, EOMES was essential for the accumulation of all NK cell subsets, especially DP that were the only ones to decrease in percentage among total NK cells in both BM and spleen compared to controls (Fig. 2C). The strong decrease in the number of immature CD11B$^-$ NK cells in the BM in the absence of EOMES indicated an essential role of this TF in early NK cell development, unlike T-BET. Of note, the paucity of NK cells in $NK\text{-}Eomes^{-/-}$ mice was not compensated by an accumulation of CD49A$^+$ILC1 either CD49B$^+$ or CD49B$^-$ (Supplementary Fig. 2A).

T-BET deficiency resulted in an accumulation of DP both in terms of percentage and number in the BM and a near lack of CD27$^-$ cells in both organs analyzed (Fig. 2C), as previously shown[4,21], confirming the essential role of this TF in terminal NK cell maturation. We also observed an inverse pattern of KLRG1 expression, which is highly expressed in terminal mature NK cells, in the absence of EOMES and T-BET, i.e., an increased percentage of KLRG1$^+$ NK cells and a decreased percentage of these cells among NK cell subsets when compared to controls respectively, suggesting that terminal maturation is promoted by T-BET and rather prevented by EOMES (Supplementary Fig. 2B). To further test this point we sorted NK cells at maturation stages 1 and 2 from $Tbx21^{-/-}$, $NK\text{-}Eomes^{-/-}$ and control mice and adoptively transferred them into un-irradiated Ly5a X B6 (CD45.1/2) mice. Transferred NK cells were allowed to acclimate for two weeks before analysis of their maturation status in the spleen. The recovery rate was low for both EOMES and T-BET deficient NK cells compared to controls (Supplementary Fig. 2C). However, recovered $NK\text{-}Eomes^{-/-}$ NK cells from either stage 1 or stage 2 were still $Eomes$-deficient (Supplementary Fig. 2D), did not differentiate into ILC1s (Supplementary Fig. 2E) and had an accelerated maturation towards the CD27-CD11B$^+$ stage 3 compared to controls, while $Tbx21^{-/-}$ NK cells had the opposite behavior (Fig. 2D), thus confirming the antagonistic effect of both TFs on NK cell maturation rate. Collectively, these results demonstrate the sequential roles of EOMES and T-BET in NK cell differentiation. Moreover, they show that in the absence of EOMES, the few developing NK cells can become hyper mature, possibly because of an excess of T-BET.

**EOMES and T-BET promote NK cell survival at different maturation stages.** To further document the role of EOMES and T-BET in NK cell development and homeostasis, we next assessed proliferation and survival of NK cell subsets developing in the absence of either factor. We evaluated NK cell proliferation using EdU incorporation or KI67 staining. The main proliferative burst in NK cells normally occurs before the acquisition of the CD11B integrin, in the BM[36]. Surprisingly, $Eomes$ deletion did not change the level of proliferation of immature CD11B$^-$ NK cells in the BM, but rather increased that of DP in this compartment. In the spleen, all $NK\text{-}Eomes^{-/-}$ NK cell subsets had an increased proliferation rate, especially CD11B- and DP (Fig. 3A and Supplementary Fig. 3A). $Tbx21$ deletion increased the proliferation rate of mature CD27$^-$ NK cells and to a lesser extent that of DP, both in the BM and spleen, while CD11B- NK cells were hardly affected.

The increased proliferation of NK cells in the periphery of both knockout mouse models could be related to the increased availability of IL-15 due to the lack of its consumption by NK cells. To address this possibility, we transferred CTV-labeled $Eomes^{-/-}$ or $Tbx21^{-/-}$ NK cells with control NK cells into Ly5a congenic mice that have normal NK cell numbers. We then monitored the percentage of divided NK cells two weeks after transfer. As shown in Supplementary Fig. 3B, $Tbx21^{-/-}$ NK cells still divided more than control NK cells in these conditions. By contrast, $Eomes^{-/-}$ NK cells had a near-normal proliferation, indicating that EOMES does not intrinsically regulate proliferation in NK cells.

To explain the reduction in NK cells upon deletion of either factor, we hypothesized that both TFs could regulate their survival. Indeed, the percentage of apoptotic Annexin-V$^+$ DP cells was increased in the absence of EOMES in both spleen and BM. Similarly, the percentage of apoptotic Annexin-V$^+$ CD27$^-$ NK cells was increased in the absence of T-BET in both BM and spleen (Fig. 3B). As IL-15 is a major mediator of NK cell survival[37], we next assessed how EOMES and T-BET regulated the response to this cytokine. Upon co-culture with IL-15 for 48 h, both EOMES and T-BET deficient NK cells showed a decreased viability compared to controls (Fig. 3C) suggesting that a balanced expression of EOMES and T-BET is necessary for optimal response to IL-15. At the subset level, the decreased viability was observed for T-BET deficient CD27$^-$ NK cells, and for EOMES-deficient DP NK cells (Supplementary Fig. 3C). CD122 surface levels were higher in T-BET deficient and lower in EOMES-deficient NK cells than in controls, respectively (Fig. 3D), showing that T-BET and EOMES regulate IL-15-mediated survival not simply through the control of CD122 expression. Altogether, these data demonstrate that EOMES and T-BET promote survival of maturing NK cells in a sequential manner, which could involve fine-tuning of IL-15 responsiveness.

**Complementary roles of EOMES and T-BET in instructing the NK cell maturation program.** Next, to gain molecular insight on the mechanisms of T-BET and EOMES action during NK cell development and maturation, we performed RNA-seq in immature CD11B- and mature CD27$^-$ NK cells from $Tbx21^{-/-}$, $NK\text{-}Eomes^{-/-}$ and control mice. In accordance with the dominant expression of EOMES in immature cells (Fig. 1), we found that the loss of EOMES but not that of T-BET had a strong impact on gene expression in immature NK cells (Fig. 4A, B and Supplementary Data 1). Inversely, in mature NK cells, T-BET had a much more prominent role than EOMES on gene transcription (Fig. 4A, B). When examining differentially expressed genes (DEGs), EOMES was more often a transactivator of gene

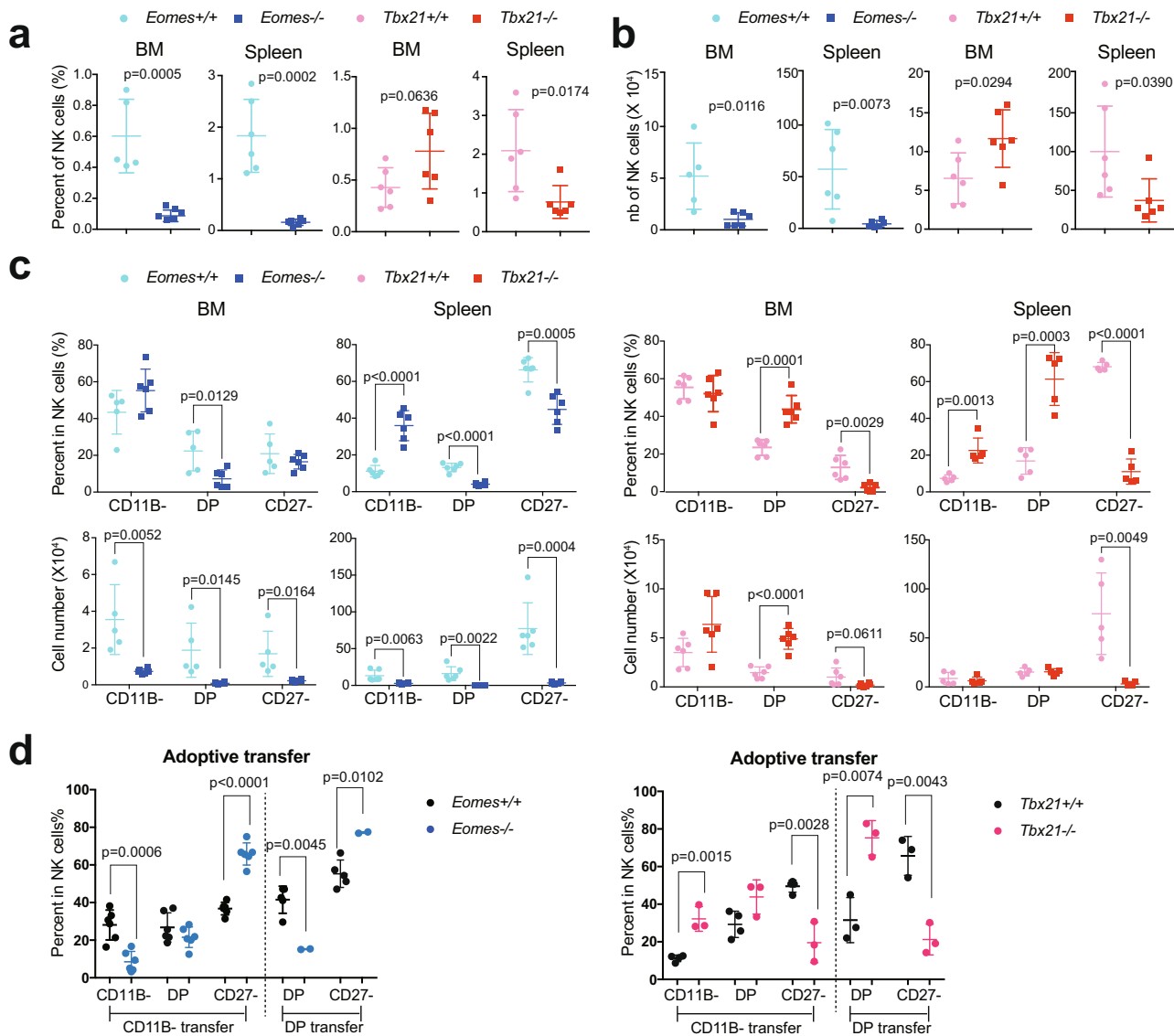

**Fig. 2 EOMES and T-BET promote NK cell survival and differentiation at different cellular transitions. a** Flow cytometry analysis of NK cell percentage in BM or spleen lymphocytes of the indicated mice. Each dot corresponds to a single mouse ($n$ = 5–6 mice in each group, pooled from 2 experiments). **b** Number of NK cells calculated from data in (**a**) combined with numeration of corresponding organs ($n$ = 5–6, pooled from 2 experiments). **c** Proportion and numbers of NK cell subsets in BM or spleen lymphocytes of the indicated mice. Each dot corresponds to a single mouse ($n$ = 5–6 mice in each group, pooled from 2 experiments). **d** NK cells of the indicated subsets (CD11B- or DP) and genotypes were FACS-sorted and then adoptively transferred into congenic unirradiated Ly5a X B6 mice (CD45.1/2). Two weeks later, spleen NK cells were purified from these mice and the CD11B/CD27 phenotype of transferred NK cells (identified by their CD45.2 expression) was analyzed by flow cytometry ($n$ = 2–6, pooled from 2 experiments). In all graphs in Fig. 2, the mean measurement ± SD is shown. Unpaired $t$ tests (two-tailed) were used for statistical analysis of data presented in this figure.

expression (84% of DEGs in immature NK cells and 52% in mature ones) while T-BET was more often a repressor of gene expression (63% of DEG in mature NK cells, Fig. 4B). We queried the Immgen database[38] to visualize the expression pattern of EOMES and T-BET-regulated genes across the whole immune system. EOMES-induced genes were found to be rather NK-specific while T-BET activated genes had a broader expression pattern (Supplementary Fig. 4A). The expression pattern of EOMES-repressed and T-BET-repressed genes was more comparable, but T-BET appeared more specialized in the repression of genes expressed in hematopoietic progenitors, or in B and T cell progenitors (Supplementary Fig. 4B). Of note, EOMES-dependent genes in immature NK cells include 23/91 of genes defining the NK cell signature as previously defined by the Immgen consortium[38,39], while only four genes were also

regulated by T-BET in this list (Fig. 4C). Inversely, in mature NK cells, 25 and 6 of these genes were regulated by T-BET and EOMES, respectively (Fig. 4C), suggesting that both EOMES and T-BET contribute to define NK cell identity, but in a sequential manner.

Overall, there was a limited overlap between T-BET and EOMES dependent genes both in immature and mature NK cells (Fig. 4D, Supplementary Data 1), with 114/166 (68%) EOMES dependent genes not regulated by T-BET and 691/744 (92%) T-BET dependent genes not regulated by EOMES. We also looked at the mutual regulation between EOMES and T-BET. There was a trend for more EOMES mRNA in the absence of T-BET and vice versa (Supplementary Fig. 4C) but this was not statistically significant, suggesting that posttranscriptional mechanisms may contribute to explain the results observed in Fig. 1 at the protein level.

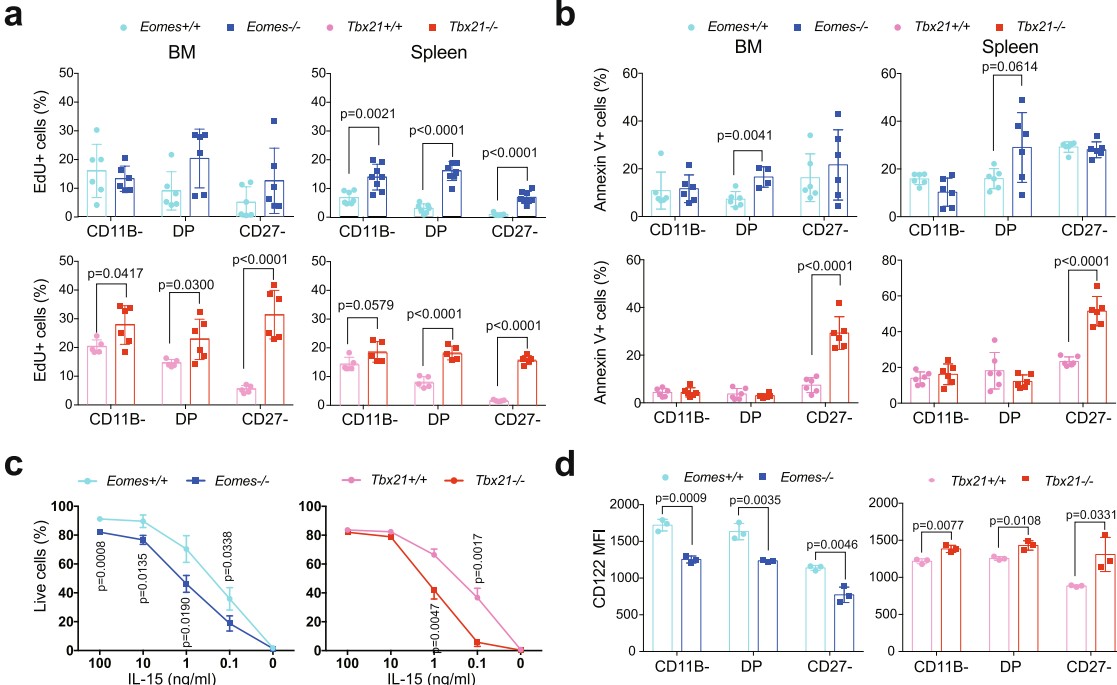

**Fig. 3 EOMES and T-BET regulate NK cell proliferation and survival at different stages. a** Mice were injected twice with EdU within a two-day-period, and EdU incorporation was measured in NK cells from the indicated organs by flow cytometry one day after the last injection. Each dot corresponds to a single mouse ($n = 6$, pooled from 2 experiments), and graphs show the mean percentage ± SD. **b** Ex vivo apoptosis of BM and spleen NK cell subsets as assessed by flow cytometry analysis of Annexin-V staining ($n = 5$–6, pooled from 2 experiments). Bar graphs show the mean percentage ± SD of Annexin-V + cells within each subset, dots correspond to individual mice analyzed. **c** Total NK cell viability. Bar graphs show the mean percentage ± SD of Annexin-V negative cells after culture of splenocytes with IL-15 for 48 h at the indicated concentrations. Data are from 3 mice representative of 2 independent experiments. **d** Flow cytometry measurement of CD122 expression in NK cells. Bar graphs show the MFI ± SD (one symbol per individual mouse analyzed). Paired t tests were used for statistical analysis of data presented in this figure. Data are from 3 mice representative of 2 independent experiments. Unpaired t tests (two-tailed) were used for statistical analysis of data presented in this figure.

Finally, 82/166 EOMES-dependent (52%) and 250/744 T-BET-dependent (35%) genes were regulated during NK cell maturation (Fig. 4E, F). Reciprocally, 294/699 genes normally regulated during NK cell maturation were dependent on either EOMES or T-BET (42%) (Fig. 4F), confirming that both factors are major drivers of NK cell maturation, acting in a complementary manner.

**Specific and shared actions of T-BET and EOMES on NK cell transcriptome.** To define which properties were commonly or specifically conferred to NK cells by EOMES and T-BET, we then performed functional annotations of the gene modules described in Fig. 4.

Only one gene was regulated together by T-BET and EOMES in both immature and mature NK cells: Cym, which encodes a mast cell enzyme of unknown function in NK cells (Fig. 4D). Fifty-two other genes were regulated both by T-BET and EOMES in at least one comparison (Fig. 4D). This gene module is presented in details in Fig. 5 and includes three subsets: genes repressed by both TFs (Fig. 5A, B), genes induced by both T-BET and EOMES (Fig. 5C–F) or genes induced by EOMES but repressed by T-BET (Fig. 5G–I).

T-BET/EOMES-co-repressed genes (Fig. 5A) notably include a cluster of genes involved in cell cycle (Top2a, Ccna2 etc), and Il7r, a hallmark gene of lymphoid progenitors and ILCs. IL7R expression was also higher at the protein level in both $Tbx21^{-/-}$ and $NK$-$Eomes$ $^{-/-}$ NK cells compared to controls (Fig. 5B).

T-BET/EOMES co-induced genes (Fig. 5C) include many genes induced by EOMES in immature NK cells and by T-BET in mature ones. This group comprises crucial mediators of granule-dependent cytotoxicity such as Prf1, Gzma and Serpinb9b, and "cell killing" was one of the significant terms in a functional annotation of this geneset (Fig. 5E) using Metascape[40]. At the protein level, we confirmed that GZMA was strongly reduced in both T-BET and EOMES deficient NK cells (Fig. 5D). Cma1 encodes for another mast cell-protease present in granules that could also be involved in cytotoxicity. We assessed the cytotoxic potential of EOMES and T-BET deficient mature NK cells using a new technique adapted to low cell numbers[41]. T-BET deficient NK cells had a normal cytotoxicity, while EOMES deficient NK cells were poorly cytotoxic compared to controls (Fig. 5F). The T-BET/EOMES co-induced gene set also included S1pr5 that is essential for NK cells egress from the BM[10,42]. To examine how T-BET and EOMES contributed to this egress, we injected T-BET and EOMES deficient mice with anti-CD45.2 Ab for 5 min, thus brightly staining only cells that were in blood sinusoids. A reduced sinusoidal fraction was observed in BM and LN NK cells from T-BET deficient but not EOMES deficient mice (Supplementary Fig. 5A), showing that even though both TFs regulate S1pr5, T-BET is the major factor promoting blood circulation of NK cells. The frequency of blood NK cells was also quantified, and NK cells were strongly reduced in the absence of both T-BET and EOMES. However, in T-BET deficient mice, NK cells were only reduced in the blood (Supplementary Fig. 5B, C) and not in the BM (Fig. 2A), which confirms the specific role of T-BET in NK cell blood circulation.

Several genes showed however opposite regulation by T-BET and EOMES (Fig. 5G), for example Cd27 and Cd69 were induced by EOMES but repressed by T-BET in NK cells, which we could also

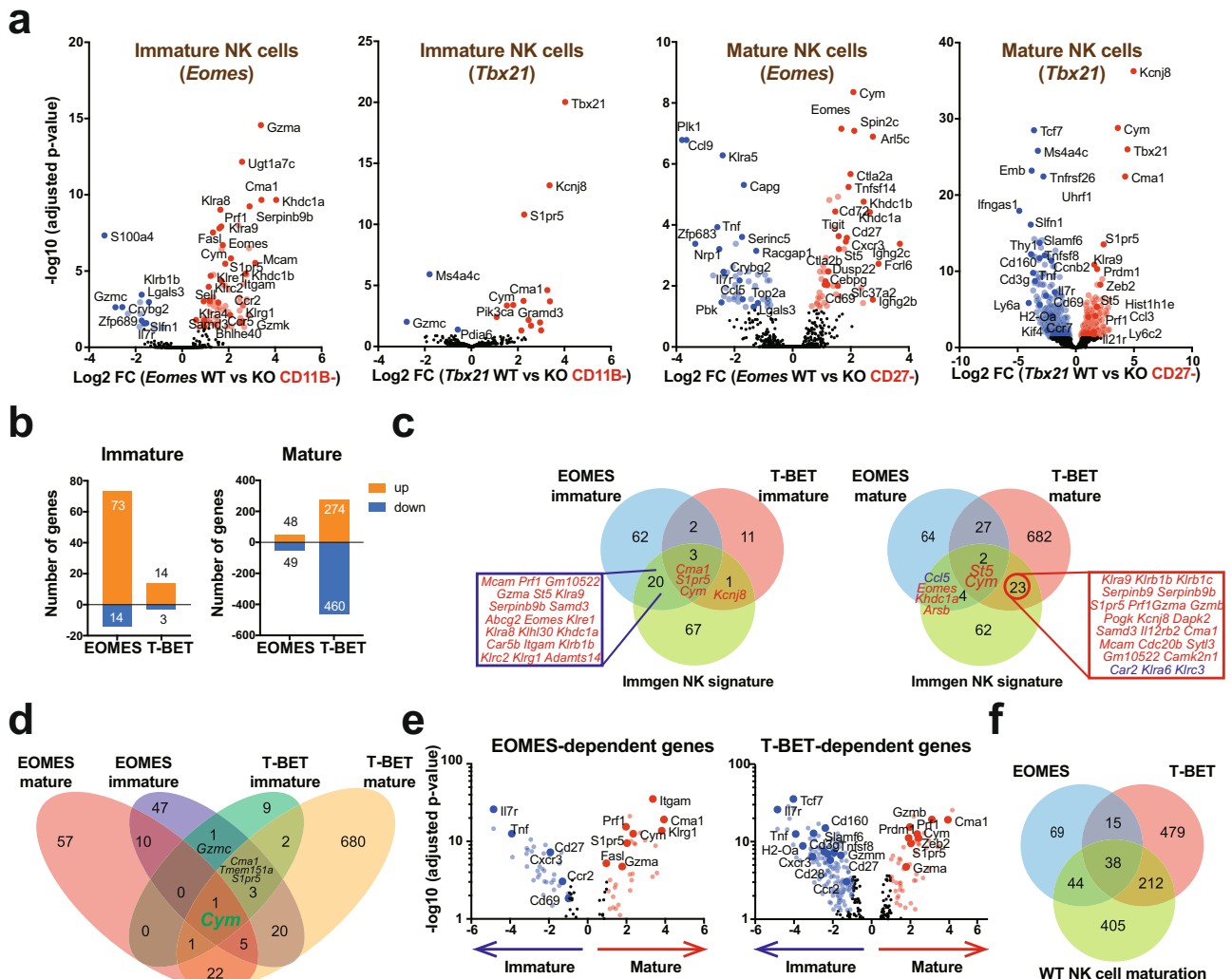

**Fig. 4 EOMES and T-BET have mostly distinct roles on NK cell transcription during maturation.** RNA-seq analysis of sorted *Tbx21*$^{-/-}$, *Eomes*$^{-/-}$ and appropriate control CD11B- and CD27$^{-}$ NK cells ($n = 3$ per group with three sorts in total). DEG were selected based on adjusted $p < 0.05$ (as determined by use of negative binomial generalized linear models). **a** Volcano plots of gene expression in immature CD11B- or mature CD27$^{-}$ NK cells comparing T-BET or EOMES deficient cells with appropriate controls. Selected genes are highlighted with their names. **b** Bar histograms showing the number of genes induced or repressed by T-BET or EOMES in immature and mature NK cells. **c** Venn diagram showing the overlap between the Immgen-defined NK cell signature and the EOMES-activated and T-BET-activated genes in immature and mature NK cells. **d** Venn diagram showing the overlap between T-BET regulated and EOMES-regulated genes in immature and mature NK cells. **e** Volcano plots comparing the expression of T-BET or EOMES dependent genes between WT immature and mature NK cells. Selected genes are highlighted with their names. DEG were selected based on adjusted $p < 0.05$ (as determined by use of negative binomial generalized linear models). **f** Venn diagram showing the overlap between T-BET-dependent, EOMES-dependent and maturation-regulated genes.

confirm at the protein level (Fig. 5H, I). These genes are normally expressed at high levels in immature NK cells and repressed in mature NK cells[26] suggesting that when T-BET is up regulated upon terminal maturation, it substitutes EOMES on the promoter of these genes and represses their expression. Other genes may be regulated in a reciprocal manner, i.e., repression by EOMES and induction by T-BET. This is the case of *Cd226* encoding for DNAM1, which expression was not statistically different between control and KOs at the mRNA level (Supplementary Data 1), but was clearly different at the protein level (Fig. 5J).

**EOMES-specific role in the induction of hallmark NK cell genes.** EOMES specific genes (i.e., regulated either in immature or in mature NK cells by EOMES and not T-BET) include mainly EOMES-inducible ones (Fig. 6A). A functional annotation of EOMES-activated genes suggested a link with NK cell activation,

cytotoxicity and IFNγ production (Fig. 6B). Indeed, this module contains a large cluster of NK cell receptors (*Klra8, Klre1, Klra4, Klrc2, Klrg1*) and the signaling adaptor SAP (encoded by *Sh2d1a*), suggesting that EOMES is specifically involved in the induction of NK cell receptors and associated signaling and functions. Indeed, when looking at the expression of a battery of NK cell receptors at the protein level, we observed a defect in the expression of several Ly49 or SLAM receptors in immature NK cells deficient for EOMES (Fig. 6C).

EOMES also activated the expression of the CD11B integrin (encoded by *Itgam*, Fig. 6D), and genes involved in the cytotoxic function such as *FasL* and *GzmK* or in trafficking such as CD62L (encoded by *Sell*, Fig. 6E) or the granule-contained CCL5 chemokine. Finally, EOMES induced the expression of many genes of unknown function such as *Khdc1a* and *Khdc1b* and several lncRNA (*Mirt1, Gm12596* etc, see Supplementary Data 1).

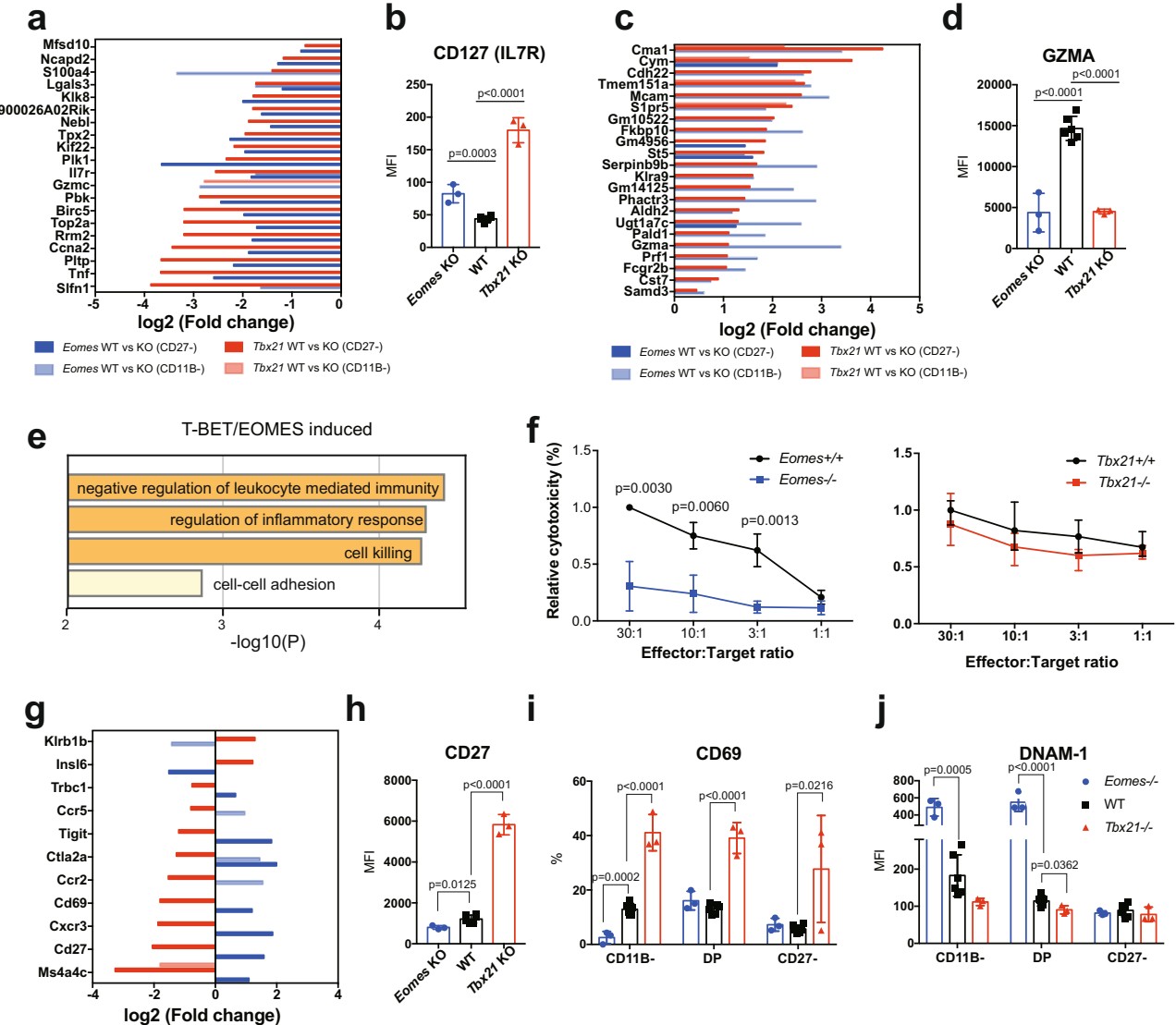

**Fig. 5 Shared transcriptional activities of T-BET and EOMES in NK cells.** RNA-seq analysis of sorted $Tbx21^{-/-}$, $Eomes^{-/-}$ and appropriate control CD11B- and CD27− NK cells ($n = 3$ per group with three sorts in total). DEGs were selected based on adjusted $p < 0.05$. **a** T-BET/EOMES co-repressed genes in immature or mature NK cells. Bar graphs show the log2 transformed fold change of genes between controls and T-BET or EOMES deficient mice, as indicated. **b** Bar graph showing the mean ± SD of IL7R expression in NK cells of the indicated genotype as measured by flow cytometry. The different controls for each genotype are pooled and annotated as "WT" in the graph which is same for the following graphs in (**d**, **h**–**j**). $N = 3$. **c** T-BET/EOMES co-induced genes in immature or mature NK cells, Bar graphs show the log2 transformed fold change of genes between controls and T-BET or EOMES deficient mice, as indicated. $N = 3$. **d** Bar graph showing the mean ± SD of GZMA expression in NK cells of the indicated genotype as measured by flow cytometry. $N = 3$. **e** Functional annotation of the T-BET/EOMES induced gene set using Metascape. Bar graphs show selected terms among the most significant ones. **f** Cytotoxicity assay using sorted $Tbx21^{-/-}$, $Eomes^{-/-}$ and appropriate control CD27− NK cells as effectors and RMA-KR-Nano-luc cells as targets. Graphs show the mean cytotoxicity ± SD. Data are from 3 mice in each group, pooled from two independent experiments. **g** Bar graph showing the log2 transformed fold change of genes regulated in opposite ways by T-BET and EOMES, between controls and T-BET or EOMES deficient mice, as indicated. $N = 3$–6. **h**–**j** Bar graphs showing the mean ± SD of CD27 (**h**), CD69 (**i**) and DNAM-1 (**j**) expression in NK cells of the indicated genotype as measured by flow cytometry. $N = 3$. Unpaired $t$ tests (two-tailed) were used for statistical analysis of data presented in this figure.

To identify putative downstream mediators of EOMES transcriptional activity, we also examined TFs whose expression was dependent on EOMES. We found that *Kcnip3, Lzts1* and *Bhlhe40* were down regulated in the absence of EOMES, either in immature or in mature NK cells (Fig. 6F). *Bhlhe40* is known to promote mitochondrial metabolism in resident memory T cells[43] and could therefore also regulate NK cell development. EOMES also behaved as a specific repressor of a subset of TFs including *Cebpg, Cbx2, Zfp689* and *Zfp683*. The latter, also known as Hobit, was previously shown to be essential for ILC1 development[44].

This suggested that $Eomes^{-/-}$ NK cells were closely related to ILC1s, even though they were gated as CD49B + CD49A−. To address this point, we performed a single cell RNA-seq analysis of CD11B− and CD11B + NK cells and of ILC1s from WT and NK-$Eomes^{-/-}$ mice. Results presented in Fig. 6G, Supplementary Fig. 6 and Supplementary Data 2 show that $Eomes^{-/-}$ NK cells cluster with WT NK cells, apart from ILC1s, thus showing that the loss of EOMES is not sufficient to induce reprogramming into ILC1s in developing NK cells.

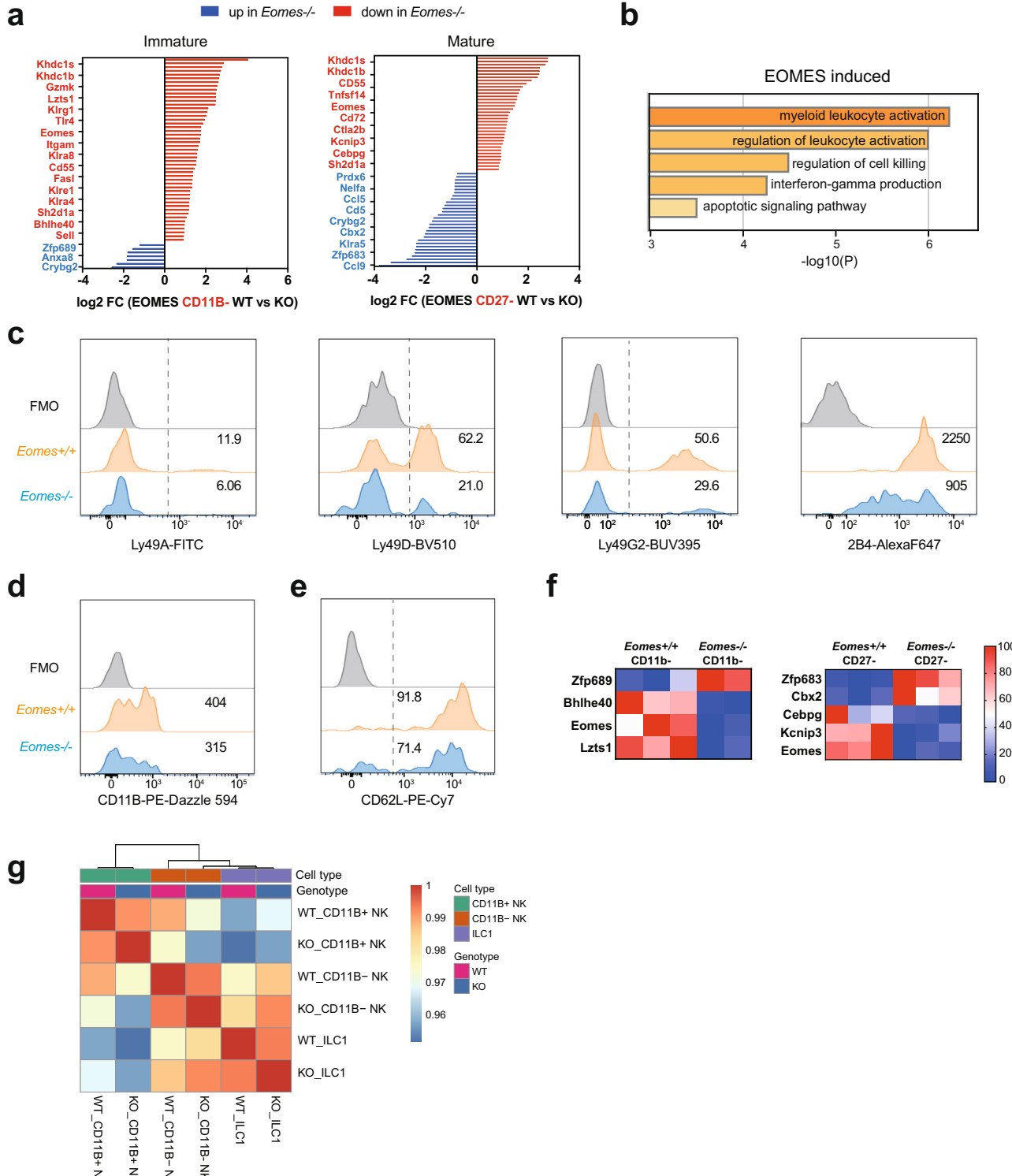

**Fig. 6 EOMES-specific role in the induction of hallmark NK cell genes.** RNA-seq analysis of sorted *Tbx21*−/−, *Eomes*−/− and appropriate control CD11B- and CD27− NK cells (N = 3 per group with three sorts in total). DEGs were selected based on adjusted *p* < 0.05. **a** Genes specifically regulated by EOMES in immature or mature NK cells. Bar graph show the log2 transformed fold change between controls and EOMES deficient NK cells, as indicated. A few selected gene names are shown. **b** Functional annotation of the EOMES induced gene set using Metascape. Bar graphs show selected terms among the most significant ones. **c–e** Overlayed FACS histograms showing the expression of the indicated surface molecules in EOMES deficient vs control immature NK cells, as indicated. Grey histograms show control staining. **f** Heatmaps showing the log2 transformed fold change in the expression of the indicated TF between control and EOMES deficient NK cells. **g** The indicated subsets were subjected to scRNA-seq analysis as detailed in the methods. The heatmap shows the correlation between averaged profiles per sample, calculated on the top 100 most variable genes.

**T-BET-specific role in the repression of pluripotency and cell proliferation**. T-BET specific genes represent a large cluster of genes, with two thirds of them being repressed by T-BET and the other ones being induced by this TF (Fig. 7A for selected genes, and Supplementary Data 1 for the complete list). A functional annotation of T-BET repressed genes retrieved "cell cycle" and related terms as the most significant ones (Fig. 7B). "Adaptive immunity", "hematopoietic cell lineage", "T cell activation" and "myeloid cell differentiation" were other significant terms, which reflects the ability of T-BET to repress progenitor/pluripotency genes or genes expressed in other immune lineages. For example, T-BET strongly repressed the expression of *Tcf7* or that of CD3 subunits. *Csf2*, encoding for GM-CSF was also specifically repressed by T-BET and not changed by EOMES.

Gene repression could be direct or indirect via other TFs induced by T-BET. Indeed, the most significant functional term associated with T-BET induced genes was "negative regulation of gene expression, epigenetic". Indeed, T-BET induced the expression of a large cluster of histone subunits (*Hist1h1c*, *Hist1h1e*, *Hist1h3a*) or histone methylation enzymes (*Phf1, Hdac5, Ezh1, Epc1*, Fig. 7C). Moreover, T-BET specifically induced the expression of many transcriptional repressors (Fig. 7C), some of which already known to regulate NK cell maturation such as *Zeb2*[26] and BLIMP (encoded by *Prdm1*)[45], and some others like *Pogk* or *Sertad1* with no described role in NK cells.

The functional analysis of T-BET-induced genes (Fig. 7B) also retrieved "IL-12 pathway" and "signaling by interleukins", which reflects the ability of T-BET to endow NK cells with specific response to cytokines. In particular, T-BET induced the expression of IL-12 and IL-21 receptors. As a consequence, *Tbx21*−/− NK cells had a defective ability to respond to IL-12 as measured by STAT4 phosphorylation (Fig. 7D), and *Tbx21*−/− but not *Eomes*−/− had an impaired IFNγ secretion in response to IL-12/18 stimulation (Fig. 7E, F). "Macroautophagy" was also associated with T-BET-induced genes (Fig. 7B) while several terms related to anabolic pathways were associated with T-BET-repressed genes. This correlates with the known role of autophagy in NK cell development and survival[46], and with the increased apoptosis observed in the absence of T-BET[19] (Fig. 3).

Importantly, even though T-BET is highly expressed in NK cells, we cannot exclude that some of the gene deregulations observed in *Tbx21*−/− NK cells could be due to a role for T-BET in the NK cell environment. To address this possibility, we generated BM chimeric mice by transplanting a 1:1 mix of BM cells from Ly5a (CD45.1) and *Tbx21*−/− mice (CD45.2) into sublethally irradiated Ly5axC57BL/6 (CD45.1/2) mice. Eight weeks after transfer, we performed a 23-parameter flow cytometry characterization of Ly5a vs *Tbx21*−/− NK cells in chimeric mice, and compared the results with those obtained when NK cells were characterized in individual donor mice (i.e., Ly5a vs *Tbx21*−/−). As shown in Supplementary Fig. 7, a strong correlation was observed between the expression ratios measured for each parameter in WT vs *Tbx21*−/− mice and those in BM chimeric mice, suggesting that most of the effect of T-BET in NK cells is cell intrinsic.

**EOMES and T-BET share most of their DNA binding sites**. To complement the RNA-seq analysis, we sought to identify T-BET and EOMES binding sites in NK cells. In an effort to decrease experimental variability and avoid a bias linked to the use of different antibodies, we generated two mouse models allowing ChIP-seq analysis of T-BET and EOMES using the same antibody. In these models, an HA-V5 encoding tag was inserted at the 3' end of *Tbx21* and *Eomes* coding sequences (Fig. 8A, see "Methods"). Flow cytometry and WB analyses validated the

expression of tagged T-BET and EOMES in NK cells even though HA-V5 tags decreased the expression of both EOMES and T-BET to some extent (Supplementary Fig. 8A, B). NK cells were present in normal frequency in both HA-V5-tagged mouse strains (Supplementary Fig. 8C), but the HA-V5 tag affected NK cell maturation in T-BET-HA-V5 mice (Supplementary Fig. 8D). The latter effect was, however, much less important than that observed in *Tbx21*−/− mice, and we therefore proceeded with ChIP-seq of EOMES and T-BET using an HA antibody in freshly isolated NK cells from both models. We verified that EOMES and T-BET could be pulled down by the HA antibody using EOMES and T-BET western blotting (Supplementary Fig. 8E). T-BET were not co-immunoprecipitated upon EOMES pull down and vice versa (Supplementary Fig. 8E) suggesting that these TFs did not interact with each other, at least in resting NK cells.

The ChIP-seq experiment was performed on two replicates of each genotype and on C57BL/6 NK cells as a control, and the peak calling was performed relative to the control cells. We controlled the similarity of both replicates for each ChIP-seq (Supplementary Fig. 9A). Both T-BET and EOMES were found to be often bound to putative promoter or intron regions and less frequently to more distant sites relative to transcriptional start sites (Fig. 8B). We found comparable numbers of binding sites for EOMES and T-BET (5682 and 4774, respectively, Fig. 8C). About half of the peaks were shared between both TFs (Fig. 8C) according to the ChIP-seq peak calling algorithm. The same proportions were kept when assigning genes to each peak using the nearest TSS method (Fig. 8D). This suggested that T-BET and EOMES compete for many of their binding sites. This conclusion was even more evident when visualizing a heatmap of the union of all EOMES and T-BET peaks (Supplementary Fig. 9B) or the mean profile of these peaks (Supplementary Fig. 9C) suggesting that most binding sites were in fact shared between T-BET and EOMES. Importantly, the control ChIP-seq yielded only background values at the position of T-BET and EOMES binding sites (Supplementary Fig. 9B, C), which validated the specificity of our ChIP-seq analysis. Moreover, EOMES and T-BET peaks identified in our study were also identified in previous studies in NK or activated CD8 T cells that used anti-EOMES or anti-T-BET antibodies (Figs. S9D, E). Finally, we scanned EOMES and T-BET ChIP-seq peaks with the FIMO tool[47] for occurrences of the T-box motif and found it in 973 EOMES peaks and 786 T-BET peaks (about 20% in each case). To confirm this point in an unbiased manner, we then performed de novo detection of TF-binding motifs under T-BET and EOMES ChIP-seq peaks using the MEME-chip motif discovery suite[48]. As expected, the canonical T-box motif was enriched under both EOMES and T-BET peaks, albeit more significantly for EOMES (Fig. 8E, F). Other motifs were also commonly enriched under EOMES and T-BET peaks, such as RREB1, RUNX1/3, SP1, and PBX3 motifs. However, there were also some EOMES-specific motifs, such as Fli1, previously identified to regulate NK cell development[49] and T-BET-specific motifs, such as STAT4, which operates downstream of IL-12 and is a known epigenetic regulator of NK cell activation[50].

We then combined the ChIP-seq analysis with our RNA-seq results in an attempt to identify direct T-BET and EOMES target genes. Genes associated with T-BET or EOMES binding were in general much more expressed than those without binding (Fig. 8G), and co-binding did not influence further this expression. Among genes for which we identified a nearby EOMES ChIP-seq peak, 79 were differentially expressed between WT and *NK-Eomes*−/− NK cells (Fig. 8H). This group of functional EOMES targets was in most cases (58/79, or 73%) induced by EOMES (Supplementary Fig. 10A and full list in Supplementary Data 3), and notably contains many genes

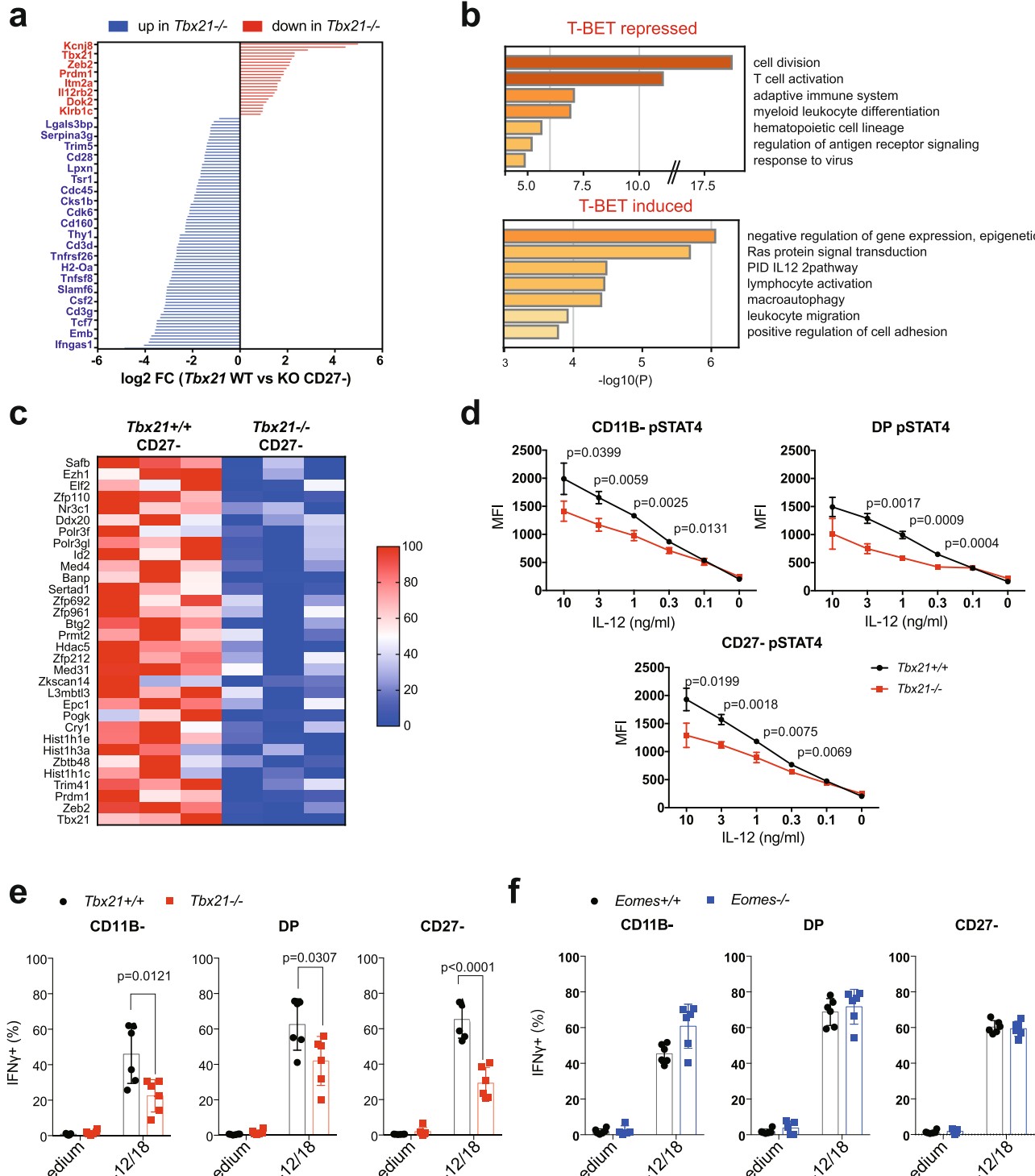

**Fig. 7 T-BET-specific role in the repression of pluripotency and cell proliferation.** RNA-seq analysis of sorted $Tbx21^{-/-}$, $Eomes^{-/-}$ and appropriate control CD11B- and CD27⁻ NK cells ($n = 3$ per group with three sorts in total). DEGs were selected based on adjusted $p < 0.05$. **a** Genes specifically regulated by T-BET in mature CD27⁻ NK cells. Bar graphs show the log2 transformed fold change between controls and T-BET deficient mice, as indicated. A few selected gene names are shown. **b** Functional annotation of the T-BET-repressed and T-BET-induced genesets using Metascape. Bar graphs show selected terms among the most significant ones. **c** Transcription factors, histone subunits or histone modifying enzymes whose expression are dependent on T-BET. The heatmap shows the log2 transformed fold change in expression between control and T-BET deficient mature NK cells. **d** Flow cytometry analysis of STAT4 phosphorylation in WT vs T-BET deficient NK cells of the indicated subset in response to stimulation with IL-12 for 1 h. Graphs show the MFI ± SD. Data are from 3 mice representative of two experiments. **e–f** Flow cytometry analysis of intracellular IFNγ expression in gated spleen NK cells of the indicated genotype following culture in medium supplemented or not with IL-12 and IL-18. Bar graphs show the mean percentage of IFNγ positive NK cells ± SD. $N = 6$ mice in 2 experiments. Unpaired $t$ tests (two-tailed) were used for statistical analysis of data presented in this figure.

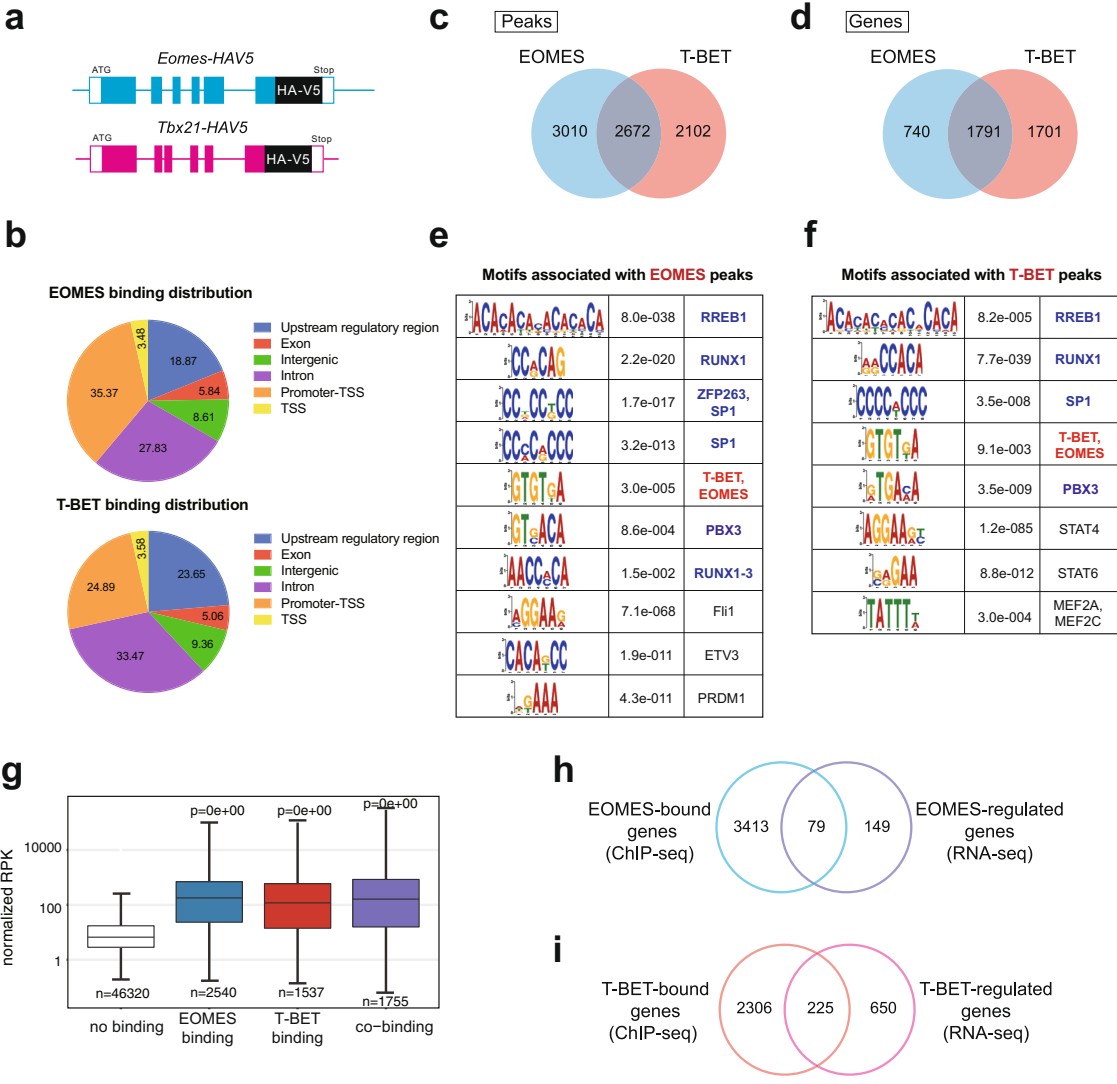

**Fig. 8 Identification of direct T-BET and EOMES targets using novel KI mice. a** Scheme of knock-in *Eomes* and *T-bet* HAV5 constructs. **b** Distribution of EOMES and T-BET peaks relative to different genomic features. "Upstream regulatory regions" refers to regions located −20 kb/−1 Kb away from the TSS, "promoter-TSS" refers to regions located −1 Kb/+100 bp around the TSS and "TSS" refers to regions located −100 bp/+1 Kb around the TSS. **c** Venn diagram showing the overlap between EOMES and T-BET ChIP-seq peaks. **d** Venn diagram showing the overlap between genes nearby EOMES and T-BET ChIP-seq peaks. **e–f** Motifs associated with (**e**) EOMES or (**f**) T-BET peaks and corresponding TFs (*E*-values are shown, not ranked). **g** Boxplot presenting expression (RNA-seq data) of genes bound by T-BET and/or EOMES (Chip-seq data). Expression of genes without any of them is shown as reference. The number of genes analyzed is indicated in the figure. Minimal and maximal values are shown. The centre gives the mean expression and the box gives the 10–90 percentile. *P* values were calculated by Wilcoxon signed-rank tests. **h** Venn diagram showing the overlap between EOMES-bound (ChIP-seq) and EOMES-regulated genes (RNA-seq). **i** Venn diagram showing the overlap between T-BET-bound (ChIP-seq) and T-BET-regulated genes (RNA-seq). In panels (**c**, **d**, **h**, **i**) *p* values were calculated through hypergeometric tests and were *p* = 0; 0; 5.61133e-40; and 6.440556e-96, respectively.

involved in cytotoxicity (*FasL, GzmA, Prf1*), NK cell receptors (*Klre1, Klrg1*), NK cell trafficking (*S1pr5, Sell*), chemo-attraction (*Cxcr3, Ccr5*) or transcription (*Bhlhe40, Prdm1*), while directly repressed genes include *Zfp683* (Hobit) and *Ptpn6* (Shp1). Examples of ChIP-seq tracks for direct EOMES targets are shown in Supplementary Fig. 10B.

Among genes for which we identified a nearby T-BET ChIP-seq peak, 225 were differentially expressed between WT and *Tbx21*−/− mature NK cells (Fig. 8I, Supplementary Fig. 10C and Supplementary Data 3). A total of 97 of them corresponded to T-BET-induced genes and notably include TF (*Zeb2, Blimp1, Id2, IRF8*), many cytokines and cytokine receptors (*IFNg, IL18r1, IL21r, IL12rb2*) but also genes involved in cytotoxicity (*FasL, Gzma, Gzmb, Prf1*) or in trafficking (*S1pr5*). Examples of ChIP-seq tracks for direct T-BET targets are shown in Supplementary Fig. 10D.

Finally, we sought to determine if EOMES and T-BET binding was associated with chromatin remodeling. To address this question, we re-analyzed a set of published ATAC-seq data for NK cell precursors (NKp), BM immature NK cells (iNK) and spleen NK cells (mature or mNK)[15]. We then compared ATAC-seq profiles at EOMES or T-BET ChIP-seq peak positions between all three conditions. Results in Supplementary Fig. 11A, B show that ATAC profiles are on average much more intense in iNK or mNK compared to NKp at these positions, indicating opening of the chromatin during NK cell differentiation at sites of EOMES or T-BET occupancy. Next, we performed a footprinting analysis of the ATAC-seq data using the TOBIAS tool[51] to identify TF binding motifs in ATAC-seq "footprints". Using TOBIAS, we performed differential TF binding analysis between NKp and iNK or NKp and mNK. For both comparisons, the most

significant differential binding was obtained for a cluster of T-box factors, that were enriched in iNK or mNK compared to NKp, EOMES and T-BET being part of this cluster (Supplementary Fig. 11C, D). When comparing iNK to mNK (Supplementary Fig. 11E), this cluster was not differential, suggesting that most of the chromatin remodeling induced by T-box factors occurs at the NKp to iNK transition, i.e., when EOMES is dominant over T-BET.

## Discussion

Here, we provide molecular insight on how EOMES and T-BET control NK cell development and maturation. Using a variety of methods, we show that EOMES is dominantly expressed in immature NK cells and that deficiency in EOMES but not T-BET strongly affects the transcriptome of immature NK cells. In addition, we found that T-BET deficiency does not affect the number of immature NK cells, while EOMES deficiency resulted in a severe decrease of this subset and of other consecutive stages. Mechanistically, our data demonstrate that EOMES is essential to specify the NK cell lineage by inducing the expression of many hallmark genes important to provide survival signals (such as NK cell receptors[52]) and by promoting optimal responsiveness to IL-15. In the absence of EOMES, immature NK cells were more apoptotic, confirming recent findings[32]. However, they did not convert into ILC1s, as shown by the scRNA-seq analysis.

We showed that T-BET was dominantly expressed in mature NK cells, and was critical for terminal NK cell maturation, which confirms previous observations[19–21,26]. We also found that EOMES and T-BET balance each other's expression at the protein level, which is important to mitigate the maturation rate. Indeed, in the absence of EOMES, residual NK cells tend to differentiate much more quickly than control NK cells. Reciprocally, the absence of T-BET resulted in the accumulation of DP NK cells. Thus, the EOMES to T-BET ratio is a crucial rheostat of NK cell maturation. Subtle changes in this equilibrium could have important consequences on the representation of maturation subsets.

T-BET acted more often as a repressor in NK cells, suppressing the transcriptional program of immature NK cells. This could be linked to its ability to induce the expression of known repressors such as Zeb2[26] and Blimp1[45], and of many enzymes, histones subunits and complexes involved in epigenetic reprogramming, as shown in the present study. We previously described that ZEB2-deficient and T-BET-deficient NK cells were phenocopies, and that ZEB2 overexpression partly compensated for T-BET deficiency in NK cells[26]. Altogether, this suggests that T-BET induction of transcriptional repressors such as ZEB2 is essential to suppress the expression of genes expressed at previous developmental stages i.e., pluripotency, progenitors or immature NK cells. Our study points to many potentially novel transcriptional regulators downstream of T-BET such as POGK that contains a Krüppel-associated box domain (KRAB) repressor domain[53], or SERTAD1, a member of the Trip-Br family of TF, known to control cell proliferation[54]. The indirect effect of T-BET via transcriptional repressors may also explain why there is overall a limited overlap between genes regulated by T-BET and EOMES, despite the fact that T-BET and EOMES DNA binding largely intersects. Interestingly, a previous analysis of EOMES and BRACHYURY actions in embryonic stem cells also showed that EOMES was rather involved in the induction of hallmark mesoderm genes, while BRACHYURY rather repressed neuroectoderm genes[55]. Thus, division of labor between different T-box TFs may operate to favor stepwise differentiation in different tissues.

EOMES regulated the expression of very few TFs, with the notable exception of Bhlhe40 and Hobit. BHLHE40 could be especially important for NK cell metabolism downstream of EOMES, as suggested by a recent study in memory CD8 + T cells[43]. In this study, Bhlhe40[−/−] memory T cells had a diminished expression of genes encoding the components of the mitochondrial membrane or genes involved in mitochondrial metabolism and/or OXPHOS, and a decreased oxygen consumption rate[43]. HOBIT has been previously reported to control tissue residency in lymphocytes, in cooperation with BLIMP1[44]. As NK cells are the only ILCs capable of circulating in the blood vasculature and the only ILCs to express EOMES, we hypothesized that EOMES promoted blood circulation by repressing Hobit. However, we found little impact of EOMES deficiency on the ability of residual NK cells to exit the BM or LN, as opposed to the effect of T-BET that was much pronounced, especially in mature NK cells. This effect could be mediated by S1PR5, which indubitably contributes to NK cell trafficking, as we previously showed[10,42]. Why ILC1s are tissue-resident and S1PR5 negative despite high levels of T-BET remains to be determined. Additional factors, such as ZEB2 that is induced at late stages of NK cell differentiation may be important to open the S1pr5 locus.

T-BET was also necessary for the survival of mature CD27[−] NK cells, which correlated with the T-BET dependent induction of the "macroautophagy" pathway as captured by the functional annotation of the T-BET dependent geneset. Autophagy is known to be absolutely essential for NK cells and more generally for all ILC types[46]. Glycolysis and oxidative phosphorylation are known to operate at very low intensity in NK cells and these pathways are even down regulated at the transcriptional level during NK cell differentiation[56]. In this context, autophagy could be especially important to preserve NK cell integrity in quiescent mature NK cells. Autophagy is negatively regulated by the mTOR pathway, and we showed that activity of both mTORC1 and mTORC2 complexes was indeed progressively decreased during NK cell maturation[56]. More recent studies have shown an interesting link between mTOR complexes and T-box factors. Indeed, deletion of Raptor, the essential subunit of mTORC1 resulted in a specific decrease of EOMES expression while deletion of Rictor had the reciprocal effect on T-BET[57]. The latter effect could be through the posttranscriptional regulation of FOXO1 that is known to negatively regulate T-BET expression in NK cells[58]. How mTORC1 specifically regulates EOMES and how the mTORC1/mTORC2 equilibrium is controlled during maturation remain open questions.

We generated novel T-BET and EOMES knock-in mice with the same HA-V5 insertion at the C-terminus of each factor. We anticipate that such tools will be very useful to study the role of both TFs in different contexts. The study of epigenetic regulation is indeed hampered by the limited availability of antibodies against TFs, and by the low expression of many of them. Using these tools, we identified a very strong overlap between EOMES and T-BET binding, which confirms a previous study in activated CD8 + T cells[59]. However, the latter study used different antibodies for T-BET and EOMES, and in vivo activated CD8 + T cells vs in vitro activated CD8 + T cells as material for T-BET and EOMES ChIP-seq respectively, which could arguably lead to technical artifacts. Our study is thus the first comparison of genome-wide DNA binding by T-BET and EOMES using untouched primary lymphocytes.

Most T-BET and EOMES peaks were not associated with differential expression of the corresponding genes in T-BET or EOMES deficient NK cells, suggesting that other factors may compensate for the lack of T-BET or EOMES binding in most cases. This point is not specific of T-box TFs as similar

conclusions have been reached for many other TFs, as previously reviewed[60]. Thus, binding is not necessarily equal to regulation, and it is probable that only a small fraction of all binding events have an important impact on gene expression.

As the EOMES/T-BET balance switches during maturation from dominance of EOMES in immature NK cells to dominance of T-BET in mature NK cells, we propose that in many occurrences and for direct T-box targets, EOMES comes first and binds T-box motifs to initiate transcription of NK cell genes. Then, when the level of T-BET increases and that of EOMES decreases, T-BET would come and replace EOMES. Even though this model requires formal demonstration, it would explain why many genes such as *Prf1* or *GzmA* are apparently co-induced by EOMES and T-BET. T-BET binding could either have the same effect as EOMES on gene expression (cases of *Prf1, Gzma, S1pr5, Il7r*, etc), or alternatively have an opposed effect (cases of *Cd27, Cxcr3, Cd69, Dnam1* etc). Opposed effects of T-BET and EOMES may involve secondary effectors as discussed above, or alternatively, different cofactors.

EOMES and T-BET were found to regulate mostly distinct gene sets, even though they bind essentially to the same genomic sites. This discrepancy could be linked to their different window of activity i.e., immature NK cells for EOMES versus mature NK cells for T-BET. Conceivably, transcriptional and epigenetic changes during maturation may indeed largely influence the outcome of TF binding. Another non-exclusive hypothesis could be that T-BET and EOMES-specific cofactors could influence how the binding of each TF influences gene transcription. The motif discovery analysis that we performed suggested that STAT4 and FLI1 could be preferential T-BET and EOMES cofactors respectively. More studies will be needed to formally address this point.

By reanalyzing a published set of ATAC-seq data we found large epigenetic changes that occur at sites of T-BET or EOMES binding during differentiation from progenitors to immature NK cells. As EOMES is dominant in immature NK cells, especially in the BM where T-BET is hardly expressed, this suggests that most epigenetic changes occurring during early NK cell specification are induced by EOMES. Moreover, we applied a novel computing tool (i.e., Tobias) that can analyse "footprints" in ATAC-seq data caused by TF binding. Using this tool, we confirmed the major impact of T-box TFs in NK cell differentiation from progenitors. Interestingly we did not uncover differential binding of T-box TFs between immature and mature NK cells, which likely reflects the strong impact of EOMES and T-BET on gene transcription in immature and mature NK cells, respectively.

In conclusion, our study reveals a major role for EOMES in NK cell lineage specification and induction of NK cell hallmark genes such as NK cell receptors and genes associated with cytotoxicity. Reciprocally, we found that T-BET promotes terminal NK cell maturation and survival via the repression of lineage and pluripotency genes indirectly via the induction of different repressors and the induction of specific properties such as responsiveness to IL-12. T-BET and EOMES binding is largely overlapping despite the specific effect of each TF on gene expression. This suggests that stage-specific epigenetic marks or specific co-factors influence the outcome of T-box binding on gene regulation. Our study also points to T-box TFs and especially EOMES as major drivers of epigenetic changes during NK cell differentiation.

## Methods

**Mice.** Wild-type C57BL/6 J mice were purchased from Charles River Laboratories (L'Arbresle, Stock No: 000664). *Ncr1-iCre*[61], *Eomes*[lox/lox] [62] and *Tbx21*[−/−] [18] mice were previously described. This study was carried out in accordance with the French recommendations in the Guide for the ethical evaluation of experiments using laboratory animals and the European guidelines 86/609/CEE. The bioethic

local committee CECCAPP and the French Ministry of research approved all experimental studies involving live mice (protocol APAFIS#29780-2021101817146456). Mice were bred and maintained under specific pathogen-free conditions in the Plateau de Biologie Expérimentale de la Souris (PBES), our animal facility. Age-matched (6-12-weeks old) and sex-matched littermate mice were used as controls. Experimental and control mice were housed and bred together. Mice were euthanized by cervical dislocation or $CO_2$ inhalation.

**Flow cytometry.** Single-cell suspensions of mouse spleen, liver, blood and BM were obtained and stained. Intracellular stainings for TFs were performed using Foxp3 kit (eBioscience). Cell viability was measured using Annexin-V (BD Biosciences)/live-dead fixable (eBiosciences) staining. Lyse/Fix and PermIII buffers (BD Biosciences) were used for intracellular staining of phosphorylated proteins. Flow cytometry was carried out on a FACS LSR II, or a FACS Fortessa (Becton–Dickinson). Data were analysed using FlowJo (Treestar). Antibodies were purchased from eBioscience, BD biosciences, R&D Systems, Abcam, Beckman–Coulter, Miltenyi or Biolegend. We used the following antibodies (Supplementary Table 1): anti-CD3 (clone 145-2C11), anti-CD19 (clone 1D3), anti-NK1.1 (clone PK136), anti-NKP46 (clone 29A1.4), anti-CD11B (clone M1/70), anti-CD27 (clone LG.7F9), anti-CD49A (clone HA31/8), anti-CD49B (clone DX5), anti-CD122 (clone SH4), anti-CD127 (clone A7R34), anti-CD62L (clone MEL-14), anti-Ly49A (clone YE1/48.10.6), anti-Ly49D (clone 4E5), anti-Ly49G2 (clone 4D11), anti-CD226 (clone 10E5), anti-Tbet (clone 4B10), anti-EOMES (clone Dan11mag), anti-Granzyme A (clone 3G8.5), anti-KI67 (clone SolA15), anti-KLRG1 (clone 2F1), anti-STAT4 (clone 38/p-Stat4), anti-HA (clone 6 E2), anti-IFNγ (clone Dan11mag).

**Cell sorting.** Murine NK cells were isolated from the spleen by magnetic cell sorting incubating for 20 min at 4 °C with a cocktail of biotin-conjugated mAb: anti-CD3ε, CD5 (53–7.3), CD19, Ly6G, F4/80, CD24, CD4, CD8, CD14 and Ter-119. The Anti-Biotin MicroBeads (20 min) were applied in addition with the DEPLETE program on the autoMACS® Separator (Miltenyi, Biotec Inc., CA, USA). A total of 50–90% pure NK cells were obtained using this procedure. Cells were then subsequently sorted into different subsets using a FACSAria Cell Sorter (Becton–Dickinson, San Jose, USA). Purity of sorted cell populations was over 98% as checked by flow cytometry.

**Immunostaining for confocal analysis.** Freshly sorted NK cells were seeded in a 96-Well Optical-Bottom Plate pre-coated with 100 μg/mL poly-L-lysin for 1 h incubation at 37 °C. After 4% PFA fixation for 15 min and three PBS washes, blocking step (PBS 3% BSA) for 30 min at RT was followed. Immunostainings were performed after a permeabilization step with 0.05% Triton X-100 for 7 min. Primary antibody antibodies were diluted in 3% BSA-PBS and added to the cell for one-hour incubation at room temperature. Primary antibodies used in this study include rat anti-CD122 (Bio X Cell, 4 μg/mL); rabbit anti-EOMES (Abcam, 4 μg/mL); FITC-conjugated mouse anti-T-BET (Biolegend, 1/50). After three washes with PBS, cells were incubated with the appropriate Alexa 555 conjugated anti-rat, AF647 conjugated anti-rabbit secondary antibody at 2 μg/mL in 3% BSA-PBS and add to the cells along with DAPI (4 μg/mL) for one-hour incubation at room temperature. After three gentle washes with PBS, cells were observed with a Zeiss LSM 800 laser scanning confocal microscope. The images and relative quantification were processed using Image J software.

**Imaging combined with flow cytometry analysis.** Sorted NK cells were prepared and stained with appropriate antibodies as described in "Flow cytometry". The well-stained cells were then analyzed by Image Stream X technology (Amnis) at magnification x40 using IDEAS software. Mean fluorescence intensity (MFI) of T-BET and EOMES were analysed by applying masks (IDEAS software) to discriminate nuclear and cytoplasmic area based on the DAPI staining.

**Measurement of in vivo cell proliferation.** Mice were given two continuous intraperitoneal injection of 0.5 mg EdU (BD Bioscience). 12 h after the last EdU injection, mice were killed and organs harvested. Cells derived from BM and spleen were stained with mAb as described in "Flow cytometry". After fixation and permeabilization, cells were stained with FITC anti-EdU antibody, according to manufacturer instructions. EdU incorporation for different cell populations was measured by flow cytometry. In some experiments Ly5a mice were injected with Cell Trace-Violet (CTV, Thermo-Fischer, used at 0.1 uM)-labelled spleen cells from C57Bl/6 mice or *Tbx21*[−/−] mice. Two weeks later, spleen NK cells were purified and CTV dilution of transferred NK cells was monitored by flow cytometry.

**T-BET and EOMES expression upon cytokine stimulation.** $2 \times 10^6$ splenocytes from C57BL/6 mice were prepared and then were cultured for 24 h with/without following cytokines IL-12 (25 ng/ml), IL-18 (5 ng/ml) and IL-15 (100 ng/ml). Cells were collected and were stained with anti-NK1.1, anti-CD3, followed by intracellular staining with anti-T-BET and anti-EOMES before analysis by flow cytometry.

**IFN-γ production and degranulation upon cytokine stimulation.** $1.5 \times 10^6$ splenocytes were cultured with Golgi-stop (BD Biosciences) in the presence of anti-CD107a for 4 h. Cytokines were used at the following concentrations: IL-12 (25 ng/ml), IL-18 (5 ng/ml). Surface and intracellular stainings were then performed and IFN-γ production as well as CD107a exposure was measured by flow cytometry.

**Adoptive transfer.** NK cells were sorted into two populations CD11B- (CD11B-CD27 + ) and DP (CD11B + CD27 + ). Purity of sorted cell populations was >98% as checked by flow cytometry. For cell transfer, $2 \times 10^4$ to $4 \times 10^5$ sorted cells were injected intravenously into unirradiated Ly5a X B6 (CD45.1/2) mice. The presence of transferred cells was analyzed at days 14 after transfer by cytofluorimetric analysis of NK cell-enriched splenocytes. The expression of CD27 and CD11B was analyzed on CD19- CD3- NK1.1 + CD49A-CD49B + cells.

**Cell cytotoxicity assay.** Sorted NK cell subsets were plated in 96-well, V-bottom plates and co-cultured for 4 h with RMA-KR target cells (MHC I deficient and Rae1b positive) expressing the Nanoluciferase (lentivirus-mediated expression) at a concentration of 100 cells/well. Different ratio of NK to target cells were used: 30:1, 10:1, 3:1, 1:1, 0:1. After NK cell killing RMA-KR-derived NanoLuc was released in the culture supernatant. NanoLuc activity in the culture supernatant thus reflects target cell lysis. The total volume in culture wells was 200 µL, and plates were centrifuged briefly for 4 min at 500 x g. 50 µL of culture supernatants was collected, and NanoLuc activity was determined by adding 50 µL of NanoLuc reagent in black, flat-bottom, 96-well plates. Bioluminescence was measured for 0.1 s with a luminometer (TECAN).

**RNA-seq.** NK cells were first purified by AutoMACS to get higher purity. And then cells were stained in combination with anti-Cd3, anti-NK1.1, anti-CD49A, anti-CD49B, DAPI and subsequently sorted into different subsets (CD11B-, DP, CD27-) using a FACSAria Cell Sorter (Becton-Dickinson, San Jose, USA). Purity of sorted cell populations was over 98% as measured by flow cytometry. RNA libraries were prepared as described[63]. Briefly, total RNA was purified using the Direct-Zol RNA microprep kit (Zymo Research) according to manufacturer instructions and was quantified using QuantiFluor RNA system (Promega). 1 µl of 10 µM Oligo-dT primer and 1 µl of 10 mM dNTPs mix were added to 0.15 ng of total RNA in a final volume of 2.3 µl. Oligo-dT were hybridized 3 min at 72 °C and reverse transcription (11 cycles) was performed. PCR pre-amplification was then conducted using 16 cycles. cDNA was purified on AmpureXP beads (Beckman Coulter) and cDNA quality was checked on D5000 screen tape and analysed on Tape station 4200 (Agilent). 3 ng of cDNA were tagmented using Nextera XT DNA sample preparation kit (Illumina). Tagged fragments were further amplified and purified on AmpureXP beads (Beckman Coulter). Tagged library quality was checked on D1000 screen tape and analyzed on Tape station 4200 (Agilent). Sequencing was performed by the GenomEast platform, a member of the "France Génomique" consortium (ANR-10-INBS- 0009), on an Illumina HiSeq 4000 sequencing machine (read length 1 x 50 nt).

**Single cell RNA-seq.** Spleen NK cells and ILC1s were FACS sorted using appropriate gates (see Supplementary Fig. 1) into the wells of a 384-well capture plate. Single cell RNA sequencing was then performed (SingleCell Discoveries, Utrecht, Netherlands) using a SORT-seq protocol[64]. Sequencing data were then processed. Low quality cells were removed based on a transcript per cell cutoff and data normalization for sequencing depth per cell and log-transformation. The most variable genes in the dataset were identified and the top 2000 was used for dimensionality reduction and clustering. Principal component analysis was done on the 2000 variable genes to identify the dimensionality in the dataset and the most relevant principal components are select for downstream analysis (dimensionality reduction and clustering). The first 8 principal components were selected for analysis, based on the final distribution of the samples in tSNE and UMAP space. Correlations between samples were calculated on the profiles averaged per sample.

**Generation of EOMES-HAV5/Knockin mice.** A homologous recombination plasmid was designed to target mouse *Eomes* gene. A targeting vector containing Homology sequences to the *Eomes* locus and an ires-GFP-loxP-tACE-Cre-PGK-gb2-neo-loxP cassette was made previously[4]. Using Red/ET cloning (Gene Bridges), we replaced the IRES-GFP and upstream STOP codon by a 2xHAV5 tag (Hemagglutinin and Parainfluenza virus 5 V/P tag) and terminal STOP codon. The rest of the cassette allows selection with neomycin in both bacteria and eukaryotic cells and is auto-excisable in male mice thanks to Cre expression under the control of the testis-specific Tace promoter and the loxP sites. JM8.A3 ES cells (C57BL/6 N) were transfected with this construct and G418-resistant clones were obtained. We checked for correct homologous recombination by PCR followed by southern blot using different probes. Chimeric mice were obtained following microinjection of ES cells into C57BL/6 blastocysts and germline transmission was monitored by PCR using different sets of primers encompassing different parts of the targeted locus. The following primers were used for genotyping the animals: Ex6-F1 and Ex6-R1 (Supplementary Table 2). The knock-in fragment size is 520 bp and the WT fragment size is 359 bp.

**Generation of Tbet-HAV5/Knockin mice.** A homologous recombination plasmid was designed to target the mouse Tbet gene. Using Red/ET cloning (Gene Bridges), we first replaced Tbet stop codon in a bacterial artificial chromosome containing Tbet genomic sequence (clone number RP24-161-F21; CHORI) by a 2xHAV5-STOP-loxP-tACE-Cre-PGK-gb2-neo-loxP cassette. We then retrieved the cassette with flanking Tbet genomic sequence spanning 3.5 kb upstream and 3 kb downstream of original stop codon. Standard procedure was used to generate genetically modified JMA8.A3 ES cells and mice (refer to *Eomes-HAV5* KI mice procedure). The following primers were used for genotyping the animals: Ex6-F2 and Ex6-R2 (Supplementary Table 2). The knock-in fragment size is 509 bp and the WT fragment size is 356 bp.

**ChIP-seq.** For sample preparation, 10 million NK cells from pooled spleens of HAV5-tagged T-BET and EOMES mice were isolated and crosslinked with 1% formaldehyde for 10 min at 37 °C. Crosslinking was stopped with 0.125 M glycine on a roller for 5 min. Cells were then washed twice with ice-cold Wash Buffer including detergent (Active motif, 53042). Flash freeze the cell pellet by immersing tubes into dry ice for 10 min. Resuspend each pellet in 5 ml ice-cold Swelling Buffer (Active motif, 53042) supplemented with protease inhibitor cocktail (PIC) and PMSF for 30 min on ice. Pellets were then suspended in 300 ul SDS buffer (0.5% SDS, 10 mM EDTA, 50 mM Tris-HCl pH8) supplemented with PIC and PMSF and incubated on ice for 10 min. Chromatin were then sheared using a Probe Sonicator device (Active motif) to obtain a fragments size range between 200 and 1000 bp. After clearance by centrifugation at 4 °C, sheared chromatin was used for immunoprecipitation of HA (4 µg, CST, catalog number 3724), or normal rabbit IgG control (4 µg, Diagenode, catalog number C15410206) incubated overnight at 4 °C. Protein G magnetic-activated beads (Active Motif, catalog number 53034) were added to each immunoprecipitation reaction and incubate for 3 h at 4 °C. Each IP were washed five times with Wash buffer (Active motif, 53042) and chromatin-antibody complexes were eluted with Elution Buffer AM4. Chromatin was then reverse cross-linked and DNA was purified according to the manufacturer's instructions (Active motif). Before sequencing, we pooled three ChIPs for the same mice. Paired-end sequencing 2 x 100 bp was performed on HiSeq 4000 (Illumina) at the GenomEast platform.

**Bio-informatic analysis.** RNA-seq. Reads were processed using an in-house RNA-seq pipeline of GenomEast facility. Briefly: raw data were preprocessed using cutadapt 1.10[65] in order to remove adaptor and low-quality sequences (Phred quality score below 20). Reads shorter than 40 bp were removed for further analysis. Remaining reads were mapped to mouse rRNA sequences using bowtie 2.2.8[66] and reads mapped to rRNA sequences were discarded for further analysis. Remaining reads were aligned to mm10 assembly of the mouse genome with STAR 2.5.3a[67]. Gene quantification was performed with htseq-count 0.6.1p1[68] using "union" mode and Ensembl 96 annotations. Differential gene expression analysis between groups of samples was performed using method implemented in the Bioconductor R package DESeq2 1.16.1[69], with the following non-default options: betaPrior = TRUE, alpha = 0.05. *P* values were adjusted for multiple testing using the Benjamini and Hochberg method.

ChIP-seq. The Encode Pipeline v1.6.1 was used to map reads and detect T-BET and EOMES peaks. Briefly, bowtie v2.3.4.3[66] was used to align reads to the reference mouse genome mm10/GRCh38. spp v1.15.5[70] was used to detect around 300,000 peaks and the IDR method was used to select reproducible peaks with an IDR threshold < = 0.05. The « IDR optimal » set of peaks was selected for downstream analysis. Controls are flag-HA chipped samples in WT NK cells. Peaks were annotated relative to genomic features using Homer annotatePeaks.pl v4.11.1[71]. Annotations were extracted from Ensembl v96.

Reproducibility of ChIPs. We compared the number of reads in detected peaks in the replicates. BEDtools makewindows v2.26.0[72] was used to compute all non-overlapping 10 Kb long bins along the mouse genome. BEDtools intersect was used to count the number of reads falling into each bin for all IP samples. Read counts per bin are presented in a scatterplot and the Spearman correlation coefficient is computed.

Heatmaps were generated using Deeptools v3.5[73] d using the tool bamCoverage to generate bigwigs files with a step of 10 nt. Bigwig files were normalized using the RPGC method. Then, the tool computeMatrix was used to generate a count matrix at the positions of interest and finally the tools plotHeatmap and plotProfile were used to generate heatmaps and mean profile plots. Data presented on the heatmap and mean profile are pooled by condition.

Motif analysis. The tool FIMO[47] of the MEME suite v4.10 was used to detect the Tbox motif EOMES/MA0800.1 (source Jaspar). The tool MEME-chip[48] was used with default parameters except for « -meme-mod zoops -meme-nmotifs 20 -meme-minw 8 -meme-maxw 25 » to de novo detect motifs in T-BET and EOMES peak sequences. We detected motifs in sequences located + /−100 nt around peak summits.

**Analysis of public data.** ChIP-seq datasets. The following published datasets were downloaded from GEO or SRA in SRA format and converted to FASTQ format using the fastq-dump program in the sratoolkit (version 2.1.9).

| Sample name | GEO/SRA accession | Publication |
|---|---|---|
| 87_T-bet_Sp_NK | GSM2056378 | Shih et al. |
| Tbet_WT_Th1 | GSM836124 | Nakayamada et al. |
| CD8_TBET_WT.1 | SRX1070596 | Dominguez et al. |
| Eomes_ChIPseq_rep1 | GSM3900380 | Wagner et al. |
| b6_cast_lcmv_cd8_d07_chip_r1_eomes | GSM3612595 | van der Veeken et al. |
| b6_cast_lcmv_cd8_d07_chip_r2_tbet | GSM3612599 | van der Veeken et al. |

Reads were mapped to the Mus musculus genome (assembly mm10) using Bowtie v1.0.0 with default parameters except for "-m 1 --strata --best".

**ATAC-seq datasets.** The following published datasets were downloaded from GEO or SRA in SRA format and converted to FASTQ format using the fastq-dump program in the sratoolkit (version 2.1.9).

| GEO identifier | Sample name | Condition |
|---|---|---|
| SRR3152814 | 9_ATAC_BM_iNK_rpt1 | iNK |
| SRR3152815 | 10_ATAC_BM_iNK_rpt2 | iNK |
| SRR3152822 | 17_ATAC_BM_NKp_rpt1 | NKp |
| SRR3152823 | 18_ATAC_BM_NKp_rpt2 | NKp |
| SRR3152848 | 43_ATAC_Sp_NK_rpt1 | Sp_NK |
| SRR3152849 | 44_ATAC_Sp_NK_rpt2 | Sp_NK |

These data were analyzed using the Encode ATAC-seq pipeline v1.5.1. Briefly, bowtie v2.3.4.3 was used to align reads to the reference mouse genome mm10/GRCh38 and MACS2 v2.2.4 was used to call OCRs. Footprinting analysis of ATAC-seq data was performed using the suite of tools TOBIAS[51]. Briefly, TOBIAS ATACCorrect corrects Tn5 insertion bias, TOBIAS ScoreBigwig calculates footprint scores within regulatory regions, TOBIAS BINDetect estimates bound/unbound transcription factor binding sites. Known motifs were from the JASPAR CORE (2020) for vertebrate non-redundant database.

**Statistical analysis.** Statistical analyses were performed using Prism 5 (Graph-Pad Software). Two tailed unpaired $t$ test, paired $t$ test, and ANOVA tests with Bonferroni correction were used as indicated in the figure legends.

**Reporting summary.** Further information on research design is available in the Nature Research Reporting Summary linked to this article.

## Data availability

ChIP-seq and RNA-seq data that support the findings of this study have been deposited in the Geo repository with the accession code GSE168242. The raw scRNA-seq data is included in this article as Supplementary data 2. The data for all graphical representations in this article are included in the Source Data File. Additional relevant information can be obtained by contacting the authors. New reagents and mouse lines are available upon request. Source data are provided with this paper.

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

## Acknowledgements

The authors acknowledge the contribution of SFR Biosciences (UMS3444/CNRS, ENSL, UCBL, US8/INSERM) facilities, in particular the Plateau de Biologie Expérimentale de la Souris, and the flow cytometry facility. The TW lab is supported by the Agence Nationale de la Recherche (ANR *GAMBLER* to TW and ANR JC *BaNK* to AM), the Institut National du Cancer, and received institutional grants from the Institut National de la Santé et de la Recherche Médicale (INSERM), Centre National de la Recherche Scientifique (CNRS), Université Claude Bernard Lyon1 and ENS de Lyon. J.Z. is the recipient of a fellowship from the China Scholarship Council (CSC). R.S. and Y.G.H. were funded by a FRM grant (AJE20161236686) to Y.G.H. L.B. is funded by the NIH research grants AI46709 and AI122217. G.D. is supported by a Research Supplement 3R01AI122217-S1 to promote diversity.

## Author contributions

J.Z., K.P., F.F., L.F., N.K., M.M., A.B., A.L.M., S.H., P.O.V., A.M. and G.D. performed experiments. S.L.G., R.S., Q.M. and M.J. performed in silico analyses. J.Z., S.L.G., A.D., L.B., Y.G.H., provided reagents, conceptual insight and helped writing the paper. T.W. wrote the paper and supervised the work.

## Competing interests

The authors declare no competing interests.
