## [Peer Review File · Nature Communications]

Reviewer #1 (Remarks to the Author):

In this study, the authors report a refined phenotyping of NK cells in the context of Eomes or Tbet deficiency. Furthermore, the authors provide a CHIP-seq resource for Tbet and Eomes binding sites. The phenotyping is well-done and the authors nicely confirm previous results showing a role for Eomes upstream of Tbx21 in regulating NK differentiation. Eomes-KO NK are more prone to terminal differentiation, though this could be due to compensatory induction of Tbet. NK from both mice are more proliferative but less sensitive to IL-15-mediated survival. Transcriptome profiling of immature and mature NK from these mice shows Tbet deficiency to have a greater defect in mature NK. The CHIP-seq data shows a large overlap between Eomes and Tbet binding sites. Other than that, the analysis is rather superficial. Overall, the manuscript is as a well-polished collection of observations concerning the role of Eomes and Tbet in NK cells. In that respect, I think it is a useful resource for the field. However, it is difficult to understand what the overall message of the study is.

Major comments:

Figure 3a. The proliferation data is difficult to interpret. Overall there seems to be a general increase in proliferation associated with deficiency of either Eomes or Tbet, except BM in Eomes KO. The authors claim a role for Eomes and Tbet in limiting NK proliferation. However, an alternative explanation could be that the dramatic reduction of total NK allows the residual NK cells which develop independently of Eomes or Tbet allows greater access to proliferation-inducing cytokines in the emptied niche. Ideally, the EdU experiment should be performed in the context of the adoptive transfer model from Figure 2D in order to show that the effect is cell-intrinsic.

Figure 3b. The Ki67 data, in several cases, does not support or refutes the EdU proliferation data. In this case, I believe EdU to be a more accurate measure of proliferation- the interpretation of Ki67 expression can vary between cell type and developmental stage. I would recommend that the Ki67 data be moved to the supplement to help clarify the main figure.

Figure 3d. Eomes or Tbet deficiency appears to reduce IL-15-mediated survival. Showing the surface expression of CD122 on Eomes^{-/-} and Tbx21^{-/-} NK cells can help develop this point.

Figure 4. Does Tbx21 or Eomes mRNA expression change in Eomes KO or Tbx21 KO, respectively?

Figure S4: The authors should quantify the peripheral NK cells in Tbet KO mice to complement their data in figure S4.

Figure 7f. Can the authors show whether Eomes deficiency has an effect on Ifng production?

Figure 8.

- 1) Although the manuscript's abstract and introduction clearly indicates the importance of genome-wide mapping to identify direct, indirect and shared Tbet/Eomes targets, the CHIP-seq data analysis within the results text is cursory and non-convincing. It requires major revision.
- 2) The lack of Tbox motifs in the Tbet pull down is troubling. It suggests the data are too poor quality to be published, or even maybe just randomly sheared chromatin that accumulates at regulatory elements. The authors should provide QC information (fraction of reads in peaks), replicate concordance, and concordance with conventional CHIP-seq.
- 3) As an extension, the authors don't show their innovative CHIP-seq approach is better, or even

equivalent to, pull-down with well characterized anti-Tbet and Eomes antibodies. In addition to concordance statistics, track snapshots for new and published data should be shown for the two assays for comparison.

4) As a further extension, it isn't clear if the ChIP-seq data were replicated. As per manuscript guidelines, replicates need to be clearly stated.

5) The authors need a no-target control to validate non-specific pull-down of easily sheared ends at active regulatory elements

6) Even with direct Chip-seq, the authors point out that many Tbet and Eomes ChIP-seq peaks do not localize to differentially expressed genes. Many of these could be artifacts from cross linking, or poor data quality. To identify direct targets, the authors should assay a form of chromatin activity (H3K27ac) or repression (H3K27me3). This, along with successful ChIP-seq would greatly raise the impact of the manuscript.

7) A concern could be that the HAV5 tag could alter normal physiological binding of the TF. There are several published CHIP-seq datasets for Eomes and Tbet in lymphocytes, including NK cells (GSE133048, GSE77695). Can the authors compare their data against publicly available CHIP-seq datasets and determine the overlap of binding sites between the data?

8) One of the conclusions made by authors is that sequential actions of Eomes and Tbet explain the different loss-of-function phenotypes. However, I'm not sure if they have shown any mechanistic evidence of this. There is a publicly available set of ATAC-seq and RNA-seq data from NK at each development stage generated by the ImmGen Consortium. The authors could integrate these data with their own to develop some of their conclusions. For example, the authors claim that Eomes and Tbet bind sequentially during maturation to facilitate NK differentiation. This claim could be tested by assembling a set of genes and associated chromatin regions that change between CD27-, DP, and CD11b+ NK in the BM and spleen and assessing the overlap of Eomes and Tbet binding with this set.

9) One conclusion made by the authors is that the role of Tbet in NK maturation is mediated through the induction of repressors such as Zeb2 and Prdm1. One possible way to show this using the current data is to show that among Tbet-bound genes, which ones are upregulated or downregulated by Tbet.

10) The author's last point about Runx3 is not clear. They just seem to claim that Eomes and Tbet bind DNA in a manner that involves Runx3 or other TFs- a point that is not expanded upon. If they wish to make this point, the authors should show some experimental evidence- perhaps CoIP- to show physical association of Runx3 or other potential TFs with Eomes and Tbet.

Minor Points:

Figure 2d. I think it is easier to understand the data if WT and KO are plotted next to each other for each population.

Figure 3b. Can the authors show the histogram or flow plot for the Ki-67 staining?

Figure S3: improve quality of the figure

Figure S8. Can the authors show statistical values (eg. Fisher's exact test) for the overlap?

Reviewer #2 (Remarks to the Author):

In the submitted manuscript by Zhang et al the authors performed an in-depth and detailed analysis of the gene targets of Eomes and T-bet in the murine natural killer (NK) cell lineage. They evaluated phenotypic and functional parameters of Eomes- and T-bet-deficient NK cells using conditional and global knockout mouse models, respectively. In addition, they performed adoptive transfer experiments to evaluate in vivo differentiation of NK cell developmental intermediates. They also performed RNAseq along with CHIP-seq experiments to identify genes directly and indirectly regulated by Eomes and T-bet within immature and mature NK cell subsets. These studies provide a wealth of useful and novel data, and in general the authors found that while there is some overlap between the genes targeted by Eomes and T-bet, these two critical transcription factors regulate distinct genes sets and in distinct phases of NK cell maturation, with Eomes working largely early in maturation and driving the expression of multiple genes including those required for NK cell cytotoxicity. In contrast, T-bet appears to have a more dominant role later during maturation with a majority of regulated genes being suppressed by T-bet.

Overall this is a wonderful study that provides many new insights into how NK cells mature in mice and how Eomes and T-bet operate to drive the NK cell phenotype. The RNAseq and CHIP-seq data will also likely serve as a very useful resource to the field. The figures are clear and easy to interpret, and the Discussion is thoughtful; indeed the notion that Eomes may be more dominant earlier during NK cell maturation with T-bet more dominant late is consistent with a number of human studies showing reciprocal expression patterns in human NK cell developmental intermediates. As far as I can tell most if not all of the findings and resource data are novel, and the authors' main conclusions are mostly supported by the presented data. This study would likely have a high impact on the field. Despite the above, I do have some comments/concerns with the submitted manuscript:

1. While the authors used a conditional knockout model to investigate the gene targets of Eomes they used a global knockout model to investigate the targets of T-bet. There is no question that T-bet is intrinsically required for adequate NK cell maturation; however, it is nonetheless possible that some of the dysregulated genes could be due to extrinsic aberrancies in the global knockout model. It would be very useful to validate the global knockout RNAseq data by performing either RNAseq (perhaps on just one NK cell subset for validation) or a more focused gene expression analysis of Tbx21^{-/-} NK cells derived in the chimeric bone marrow model in which the most of the non-NK cells in the system are genetically wild type.
2. On page 6 of the results the authors conclude that Eomes and T-bet repress one another based on the reciprocal patterns of expression and the observations that in the absence of one the other is more highly expression. However, these data are descriptive and only demonstrate correlations; can the authors further support their claims that Eomes and T-bet suppress one another?
3. The connection between NK cells and ILC1s is still not entirely clear in both mice and humans. Eomes expression appears to be a defining feature of NK cells, yet in the absence of Eomes the T-bet⁺ CD49a⁻CD49b⁺ cells are still considered to be NK lineage cells. In light of this, it would be very useful to determine how Eomes^{-/-} NK cells compare to ILC1s in terms of global gene expression patterns. In addition, did any of the adoptively transferred Eomes^{-/-} NK cells differentiate into liver ILC1s (Figure 2D)?
4. In Figure 6C-E it is not clear if the histograms show data from total NK cells or from a specific

subset according to CD27 and CD11b expression. One presumes total NK cells were tested, given that CD11b is evaluated in Figure 6D, but it would be useful to specify in the legend and text.

5. The authors created novel HAV5-tagged Eomes and T-bet mouse lines in order to perform ChIP-seq experiments. The HA and V5 blots convincingly show discrete bands for the Tbx21-HAV5 lysates in the Western blot in Figure S5B; however the bands in the Eomes-HAV5 line are more diffuse and less clear. It would be useful to additionally probe the wild-type and HAV5-tagged derived lysates using anti-T-bet and anti-Eomes antibodies in order to further validate expression.

6. Figure 8G refers to genes that were bound by T-bet and also differentially expressed between wt and Tbx21^{-/-} NK cells. Were most of these genes repressed or induced by T-bet?

7. It would be very informative and likely further significantly impact the field to know if T-bet and Eomes can physically interact given the evidence of overlapping binding sites in the ChIP-seq experiments.

Reviewer #3 (Remarks to the Author):

Zhang and colleagues study the unique or redundant roles for the transcription factors Tbet and Eomes in mouse NK cell differentiation. Previous genetic ablation studies have demonstrated that both Tbet and Eomes are required for normal NK cell homeostasis and differentiation into functionally mature NK cells. Nevertheless, how these factors work independently or in synergy to promote NK cell development is not fully understood. The authors use a series of techniques to better characterize NK cells in mice lacking Tbet or lacking Eomes in NKp46⁺ cells. These include in depth FACS-based approaches as well as genome wide transcriptomic and epigenetic studies. Together, this results suggest a complex transcriptional regulation mediated by these two related factors that includes both redundant as well as unique roles.

The data presented in Figure 1A suggests that gene dosage affects Tbet and Eomes protein expression differentially. For example, Tbet heterozygous mice have half the expression levels of Tbet protein whereas this is not the case for Eomes heterozygotes. The authors should provide statistical analysis comparing protein levels in WT versus heterozygous mice and discuss this point more fully. The Imagestream analysis (Fig 1D) shows three representative cells. The authors should provide quantification and statistical analysis of a larger number of cells to support this point. The data in Fig 1D shows inverse relationship of nuclear Tbet and Eomes in different NK cell subsets (derived from confocal studies); it is not clear how many cells are analyzed for each subset (one cell, many cells with average MFI???) and why subsets are connected with lines. This requires explanation. The co-localization of Tbet and Eomes (Fig 1E) requires correlation analysis that should be provided.

The results in Fig 2A-C are largely consistent with previous published and more recent reports (ref 32) that Eomes plays a key role in CD11b⁻ to DP stage (and beyond). The adoptive transfer studies (Fig 2D) however show that Eomes KO CD11b⁻ cells progress to DP and CD27⁻ stages. The authors interpret this as 'accelerated maturation' but this seems at odds with the data in Fig 2A-C as CD27⁻ NK cells are not present in the absence of Eomes. The authors analyze mice two weeks after adoptive transfer; how many cells are recovered? Are these cells Eomes deficient?

Data in Fig 3 suggest Tbet and Eomes suppress proliferation and control IL-15 responses. Are any particular NK cell subsets more or less susceptible with respect to survival in different IL-15 concentrations? It would be useful to know the phenotype of NK cells that survive in the presence of IL-15 in WT versus Tbet KO versus Eomes KO.

The datasets in Fig 4-7 provide a rich resource for how Tbet and Eomes regulate NK cell signatures and functions. Several targets are confirmed at protein or functional levels. It is somewhat surprising that Eomes repression of cell cycle or proliferation was not revealed by this analysis, considering the data in Fig 3. Further discussion of this point is warranted.

The novel mouse models provided by tagging Tbet and Eomes loci with HA-V5 sequences are quite interesting (Fig 8). As these are the first description of these mice, it would be important to have a better characterization of them with respect to NK cell development. Some analysis of the impact (if any) of the observed changes in Tbet and/or Eomes expression on NK cell phenotypes, homeostasis and function should be included. The number of binding sites was quite large for both factors and the peak profiles look quite similar for the examples shown in the supplemental figures. One wonders about the specificity of the antibodies used. Have the authors performed control ChIP-seq on WT mice? It is interesting that canonical binding motifs were not the same in Eomes and Tbet peaks; did these profiles change if one considers whether the peaks were Eomes-specific, Tbet-specific or shared?

Response to reviewers' comments

We thank the reviewers for their positive comments on our manuscript and for the excellent points they raised. Please find below a point-by-point response to these comments. Many novel figure and supplementary figure panels have been included in the revised manuscript. We also highlighted in yellow the changes in the text.

Reviewer #1 (Remarks to the Author):

1 - Figure 3a. The proliferation data is difficult to interpret. Overall there seems to be a general increase in proliferation associated with deficiency of either Eomes or Tbet, except BM in Eomes KO. The authors claim a role for Eomes and Tbet in limiting NK proliferation. However, an alternative explanation could be that the dramatic reduction of total NK allows the residual NK cells which develop independently of Eomes or Tbet allows greater access to proliferation-inducing cytokines in the emptied niche. Ideally, the EdU experiment should be performed in the context of the adoptive transfer model from Figure 2D in order to show that the effect is cell-intrinsic. We agree with the reviewer that the increased IL-15 availability related to the paucity of IL-15 consuming cells could drive increased proliferation of the remaining NK cells. To address this possibility, we labeled spleen NK cells from control and Tbet KO or Ncr1-Cre x Eomes lox/lox mice with CTV and transferred them into Ly5a mice (that are fully replete in NK cells) and monitored their proliferation 2 weeks after transfer. Results (now included in the manuscript in Figure S3B) show that the increased NK cell proliferation in Tbet KO mice is due to a cell-intrinsic role of this TF in the repression of cell proliferation, while the role of Eomes is cell-extrinsic, ie in the presence of a normal NK cell compartment, there is a normal or even decreased proliferation of Eomes KO NK cells. Appropriate changes have been made in the text to acknowledge this point.

2 - Figure 3b. The Ki67 data, in several cases, does not support or refutes the EdU proliferation data. In this case, I believe EdU to be a more accurate measure of proliferation- the interpretation of Ki67 expression can vary between cell type and developmental stage. I would recommend that the Ki67 data be moved to the supplement to help clarify the main figure.

We moved the Ki67 to Figure S3A as requested.

3 - Figure 3d. Eomes or Tbet deficiency appears to reduce IL-15-mediated survival. Showing the surface expression of CD122 on Eomes^{-/-} and Tbx21^{-/-} NK cells can help develop this point.

Following the reviewer's recommendation, we measured CD122 levels in NK cells from the different mouse genotypes. Results (now included in Figure 3D) show that the CD122 surface staining is decreased in Eomes deficient NK cells and increased in Tbet deficient NK cells compared to controls, respectively. These data have been commented in the results section.

4 - Figure 4. Does Tbx21 or Eomes mRNA expression change in Eomes KO or Tbx21 KO, respectively?

This question was also asked by reviewer 3 and we now provided the read counts for Tbet and Eomes in the RNAseq analysis in Figure S4C. While there was a trend

towards more Eomes in the absence of T-bet and vice versa the difference was not statistically significant. We can only speculate that additional post-transcriptional mechanisms may contribute to the difference in protein expression observed in Figure 1 between NK cells of the different genotypes.

Figure S4: The authors should quantify the peripheral NK cells in Tbet KO mice to complement their data in figure S4.

We thank the reviewer for this suggestion. We quantified the percentage of NK cells of each subset in T-bet KO, NCr1-Cre x Eomes lox/lox and control mice. The results (new Figure S5-BC) show a strong decrease of this percentage in both T-bet and Eomes deficient mice compared to control mice. However, in Tbet KO mice, NK cells were only reduced in the blood and not in the bone marrow, which confirms the specific role of Tbet in the blood circulation of NK cells. This has been discussed in the text.

Figure 7f. Can the authors show whether Eomes deficiency has an effect on Ifng production?

We now provided these data in Figure 7F, as requested by this reviewer. Eomes-deficient NK cells had a normal IFN γ secretion in response to IL12/18 stimulation, thus highlighting the specific role of T-bet in the regulation of this cytokine.

Figure 8.

1) Although the manuscript's abstract and introduction clearly indicates the importance of genome-wide mapping to identify direct, indirect and shared Tbet/Eomes targets, the ChIP-seq data analysis within the results text is cursory and non-convincing. It requires major revision.

We agree with the reviewer that our ChIP-seq analysis was not optimal, and that the data were not convincing. To improve this part of the paper, we collaborated with a bioinformatician with longstanding expertise on genomic and epigenetic analysis. She re-analyzed all ChIP-seq data and the results are now hopefully much more convincing. The changes made to the manuscript are discussed in the following points.

2) The lack of Tbox motifs in the Tbet pull down is troubling. It suggests the data are too poor quality to be published, or even maybe just randomly sheared chromatin that accumulates at regulatory elements. The authors should provide QC information (fraction of reads in peaks), replicate concordance, and concordance with conventional ChIP-seq.

As mentioned above, we reanalyzed our ChIP-seq data with the help of a trained bioinformatician. We changed the peak calling parameters and used the control ChIP (C57BL/6 NK cells) instead of the input DNA to identify Eomes and T-bet peaks. This more stringent analysis resulted in a lower number of peaks (Figure 8B-C), but downstream analyses demonstrated that they correspond to authentic T-bet and Eomes binding sites. Indeed, i) our QC analysis showed excellent concordance between replicates (figure S9A); ii) the T-box motif was found in

about 20% of peaks when scanning the sequences with FIMO, but more importantly, it was discovered de novo (ie without bias) when using the MEME-suite tools (Figure 8E-F); iii) our ChIPseq results were largely comparable to those of previously published studies that used anti-T-bet or anti-Eomes antibodies for ChIP (Figure S9D-E). Thus we are now confident that the quality of our ChIP data is overall very good considering that we used freshly isolated resting primary cells for the experiment.

3) As an extension, the authors don't show their innovative ChIP-seq approach is better, or even equivalent to, pull-down with well characterized anti-Tbet and Eomes antibodies. In addition to concordance statistics, track snapshots for new and published data should be shown for the two assays for comparison.

We thank the reviewer for this suggestion. As mentioned in the previous comment, we now showed a comparative analysis of our ChIP-seq results with previously published data obtained with NK or CD8 T cells and anti-Tbet or anti-Eomes antibodies (Figure S9-DE). This comparative analysis shows overall that most peaks identified in our study were also found in other studies, even though the quality of the pull-down is variable between studies. Looking at the overlay between mean profiles of peaks, our Eomes ChIP-seq strategy appeared better than previous studies, while the T-bet one was average.

4) As a further extension, it isn't clear if the ChIP-seq data were replicated. As per manuscript guidelines, replicates need to be clearly stated.

This important information is now provided in the text (two replicates each, see also figure S9A), and we apologize for not including it in the previous version of the manuscript.

5) The authors need a no-target control to validate non-specific pull-down of easily sheared ends at active regulatory elements

Again, this important information is now included in the manuscript (see eg Figure S9B). As mentioned above, the no-target control was even used as the reference for the peak-calling strategy, which is more stringent than most published studies (that usually use input DNA as control).

6) Even with direct Chip-seq, the authors point out that many Tbet and Eomes ChIP-seq peaks do not localize to differentially expressed genes. Many of these could be artifacts from cross linking, or poor data quality. To identify direct targets, the authors should assay a form of chromatin activity (H3K27ac) or repression (H3K27me3). This, along with successful ChIP-seq would greatly raise the impact of the manuscript.

As we previously mentioned in the discussion, a modest concordance between ChIP-seq and RNA-seq results is not specific to our study but inherent to the ChIP technique. In their previous review on how TFs regulate gene expression, Slattery and colleagues¹ stated that

“One surprising finding from early genome-wide ChIP studies was that TF binding is widespread, with thousands to tens of thousands of binding events for many TFs. These numbers did not fit with existing ideas of the regulatory network structure, in which TFs were generally expected to regulate a few hundred genes, at most.

Binding is not necessarily equivalent to regulation, and it is likely that only a small fraction of all binding events will have an important impact on gene expression”.

Thus, we don't think that the weak overlap between the RNAseq and the ChIP-seq analysis is a technical issue of our study, but rather that it reflects the natural properties of TFs.

7) A concern could be that the HAV5 tag could alter normal physiological binding of the TF. There are several published CHIP-seq datasets for Eomes and Tbet in lymphocytes, including NK cells (GSE133048, GSE77695). Can the authors compare their data against publicly available CHIP-seq datasets and determine the overlap of binding sites between the data?

As mentioned above, we followed the reviewer's recommendation and now showed a heatmap analysis comparing Eomes or Tbet binding between our study and previous other studies (Figure S9-DE). We preferred to show heatmaps instead of Venn diagrams as the overlap in the latter is always very subjective, and depends on the stringency of the parameters of the peak calling analysis.

8) One of the conclusions made by authors is that sequential actions of Eomes and Tbet explain the different loss-of-function phenotypes. However, I'm not sure if they have shown any mechanistic evidence of this. There is a publicly available set of ATAC-seq and RNA-seq data from NK at each development stage generated by the ImmGen Consortium. The authors could integrate these data with their own to develop some of their conclusions. For example, the authors claim that Eomes and Tbet bind sequentially during maturation to facilitate NK differentiation. This claim could be tested by assembling a set of genes and associated chromatin regions that change between CD27-, DP, and CD11b+ NK in the BM and spleen and assessing the overlap of Eomes and Tbet binding with this set.

We thank the reviewer for this suggestion. We downloaded and re-analyzed Immgen's ATAC-seq data for NK cell precursors (NKp), bone marrow immature NK cells, and spleen mature NK cells (mNK), that were originally generated in Shih et al's paper². We then compared the normalized ATAC counts at Eomes and Tbet peak positions between NKp, iNK and mNK. These data are now shown in Figure 9A-B and show an opening of chromatin at sites occupied by Tbet and Eomes at the NKp→iNK transition. We next performed a footprinting analysis of ATACseq data using the newly described TOBIAS tool³. This in silico analysis showed a highly significant increase in T-box factors binding between NKp and iNK, confirming the essential role of T-box factors Tbet and Eomes in specifying the NK cell lineage (Figure 9C-D). Altogether, this study suggests that Eomes (which is dominant in immature NK cells) is likely involved in the opening of chromatin in developing NK cells, and that Tbet could sustain this effect in mature cells. Regarding the switch between Eomes and Tbet, this point remains difficult to make using ChIP-seq data considering that the Eomes and Tbet datasets are overall very similar. We clarified in the text that this was just a working model and that additional studies were necessary to validate it.

9) One conclusion made by the authors is that the role of Tbet in NK maturation is mediated through the induction of repressors such as Zeb2 and Prdm1. One possible way to show this using the current data is to show that among Tbet-bound genes, which ones are upregulated or downregulated by Tbet.

We thank the reviewer for this suggestion. We now showed the list of T-bet or Eomes-bound genes for which there is a decreased or increased expression in the corresponding knockout NK cells (Table S2). Interestingly Zeb2, Prdm1, Id2 are all valid targets of T-bet (and not Eomes, except for Prdm1), suggesting that most of the repressive effect of T-bet is through these TF.

10) The author's last point about Runx3 is not clear. They just seem to claim that Eomes and T-bet bind DNA in a manner that involves Runx3 or other TFs- a point that is not expanded upon. If they wish to make this point, the authors should show some experimental evidence- perhaps CoIP- to show physical association of Runx3 or other potential TFs with Eomes and Tbet.

We agree with the reviewer that we don't have many elements in our manuscript to make a strong point on Runx3. We tried to IP T-bet and Eomes in NK cells (see response to Reviewer#2, point 7) but we failed to detect T-bet-Eomes interactions, or interaction of any of these TFs with Runx3 (data not shown). As discussed with Reviewer#2, our technical conditions might not be optimal to detect these interactions, and we therefore decided not to include these data in the manuscript, and to tune down our conclusions on Runx3.

Minor Points:

Figure 2d. I think it is easier to understand the data if WT and KO are plotted next to each other for each population.

We changed the order of this panel, as requested

Figure 3b. Can the authors show the histogram or flow plot for the Ki-67 staining?
We now showed representative FACS histograms of Ki67 staining in Figure S3A.

Figure S3: improve quality of the figure

This figure has been automatically generated from the Immgen website, and we can't really improve the quality of the graphics. But we replaced the legend to improve the reading (now figure S4).

Figure S8. Can the authors show statistical values (eg. Fisher's exact test) for the overlap?

For this figure and other Venn Diagrams (Figure S10), p-values were calculated using hypergeometric tests and mentioned in the figure legends.

Reviewer #2 (Remarks to the Author):

In the submitted manuscript by Zhang et al the authors performed an in-depth and detailed analysis of the gene targets of Eomes and T-bet in the murine natural killer (NK) cell lineage. They evaluated phenotypic and functional parameters of Eomes- and T-bet-deficient NK cells using conditional and global knockout mouse models, respectively. In addition, they performed adoptive transfer experiments to evaluate in vivo differentiation of NK cell developmental intermediates. They also performed RNAseq along with ChIP-seq experiments to identify genes directly and indirectly regulated by Eomes and T-bet within immature and mature NK cell subsets. These studies provide a wealth of useful and novel data, and in general the authors found

that while there is some overlap between the genes targeted by Eomes and T-bet, these two critical transcription factors regulate distinct genes sets and in distinct phases of NK cell maturation, with Eomes working largely early in maturation and driving

the expression of multiple genes including those required for NK cell cytotoxicity. In contrast, T-bet appears to have a more dominant role later during maturation with a majority of regulated genes being suppressed by T-bet. Overall this is a wonderful study that provides many new insights into how NK cells mature in mice and how Eomes and T-bet operate to drive the NK cell phenotype. The RNAseq and ChIP-seq data will also likely serve as a very useful resource to the field. The figures are clear and easy to interpret, and the Discussion is thoughtful; indeed the notion that Eomes may be more dominant earlier during NK cell maturation with T-bet more dominant late is consistent with a number of human studies showing reciprocal expression patterns in human NK cell developmental intermediates. As far as I can tell most if not all of the findings and resource data are novel, and the authors' main conclusions are mostly supported by the presented data. This study would likely have a high impact on the field. Despite the above, I do have some comments/concerns with the submitted manuscript:

1. While the authors used a conditional knockout model to investigate the gene targets of Eomes they used a global knockout model to investigate the targets of T-bet. There is no question that T-bet is intrinsically required for adequate NK cell maturation; however, it is nonetheless possible that some of the dysregulated genes could be due to extrinsic aberrancies in the global knockout model. It would be very useful to validate the global knockout RNAseq data by performing either RNAseq (perhaps on just one NK cell subset for validation) or a more focused gene expression analysis of Tbx21^{-/-} NK cells derived in the chimeric bone marrow model in which the most of the non-NK cells in the system are genetically wild type.

We agree with the reviewer that a conditional T-bet knockout mouse would have been better than the full KO mouse. Unfortunately, we don't have these mice and it would take very long to import and breed them in our mouse facility. In order to address this valid comment, we generated bone marrow chimeric mice (1:1 ratio of WT and T-bet KO), and we performed a FACS analysis of 23 surface and intracellular parameters. We then plotted the ratio observed for each of these parameters between WT and KO cells in BM chimeric mice as a function of the same ratio observed between NK cells of individual mice. In general, we observed a very good correlation between expression ratios in BM chimeric vs individual mice, suggesting that most of the effect of T-bet is cell intrinsic. These data have now been added in figure S7 and described in the manuscript.

2. On page 6 of the results the authors conclude that Eomes and T-bet repress one another based on the reciprocal patterns of expression and the observations that in the absence of one the other is more highly expression. However, these data are descriptive and only demonstrate correlations; can the authors further support their claims that Eomes and T-bet suppress one another?

We agree with the reviewer that alternative possibilities may explain the increase of each T-box TF in the absence of the other one. In particular, increased binding (in the absence of competition with the other T-box member) to DNA could stabilize protein expression without changing mRNA levels. In an effort to bring some

elements, we added in the manuscript the read counts for Eomes and T-bet in the RNAseq analysis (Figure S4C). They show a trend towards more Eomes in the absence of T-bet and vice versa but this was not statistically significant. We tried to overexpress T-bet and Eomes in IL-2 activated NK cells in vitro, but we failed to express both TF at significantly higher levels than the endogenous one (data not shown). Thus, we decided to tune down our conclusion that both TF could repress each other and added the alternate possibility that post-transcriptional mechanisms may explain this apparent mutual repression. We made appropriate changes in the abstract, results, and discussion. We would like to stress that these changes do not significantly alter the main messages of the article.

3. A. The connection between NK cells and ILC1s is still not entirely clear in both mice and humans. Eomes expression appears to be a defining feature of NK cells, yet in the absence of Eomes the T-bet⁺ CD49a-CD49b⁺ cells are still considered to be NK lineage cells. In light of this, it would be very useful to determine how Eomes^{-/-} NK cells compare to ILC1s in terms of global gene expression patterns.

We thank the reviewer for suggesting this important experiment. We performed a single-cell RNAseq experiment to compare at a global scale WT ILC1s and NK cells with Eomes deficient NK cells. The results of this experiment are now provided in Figures 6G and S6 and show that Eomes deficient NK cells are more similar to WT NK cells than to WT ILC1s.

3. b. In addition, did any of the adoptively transferred Eomes^{-/-} NK cells differentiate into liver ILC1s (Figure 2D)?

We performed this important control experiment. None of the transferred NK cells became CD49a⁺ (ie ILC1s), irrespective of their genotype. This information is now added in the manuscript in Figure S2E.

4. In Figure 6C-E it is not clear if the histograms show data from total NK cells or from a specific subset according to CD27 and CD11b expression. One presumes total NK cells were tested, given that CD11b is evaluated in Figure 6D, but it would be useful to specify in the legend and text.

These figure panels show data from immature NK cells, and this has now been clarified in the figure legend. CD11b expression is in fact not negative in immature NK cells, but rather low.

5. The authors created novel HAV5-tagged Eomes and T-bet mouse lines in order to perform ChIP-seq experiments. The HA and V5 blots convincingly show discrete bands for the Tbx21-HAV5 lysates in the Western blot in Figure S5B; however the bands in the Eomes-HAV5 line are more diffuse and less clear. It would be useful to additionally probe the wild-type and HAV5-tagged derived lysates using anti-T-bet and anti-Eomes antibodies in order to further validate expression.

To address this reviewer's point, we performed WB analysis of HA, Eomes and T-bet expression in total spleens of WT, T-bet-HAV5 and Eomes-HAV5 mice. Results are now shown in Figure S8B (in place of previous S5B). We consistently observed two bands for Eomes, both in WT and Eomes-HAV5 mice. Moreover, the HAV5 insertion increased the size of both T-bet and Eomes, as shown on the WB.

6. Figure 8G refers to genes that were bound by T-bet and also differentially

expressed between wt and Tbx21^{-/-} NK cells. Were most of these genes repressed or induced by T-bet?

As requested by Reviewer#1, we now refined our ChIP-seq analysis. Even though the results are not fundamentally different, the lists of genes are slightly different. We now provided (Table S2) the lists of genes bound by Eomes or T-bet and that are either induced or repressed by each TF.

7. It would be very informative and likely further significantly impact the field to know if T-bet and Eomes can physically interact given the evidence of overlapping binding sites in the ChIP-seq experiments.

Indeed, this was an interesting possibility. We addressed this question by performing anti-HA immunoprecipitations in T-bet-HAV5 and Eomes-HAV5 spleen lysates and then WB of T-bet and Eomes. The results presented in Figure R1 for the reviewer's appreciation showed that the HA antibody resulted in a good immunoprecipitation of both Eomes and T-bet, but that the untagged factor (ie T-bet in Eomes-HAV5 and Eomes in T-bet-HAV5) was not co-immunoprecipitated. This suggests that T-bet and Eomes do not interact with each other. We did not include these data in the manuscript as we cannot exclude that T-bet and Eomes may interact in some conditions that we did not test, but we could include them if the reviewer specifically requests it.

Figure R1: Eomes and T-bet do not interact with each other in resting NK cells. Spleen cells from WT, Eomes-HAV5 and T-bet-HAV5 were lysed and the lysate was subjected to anti-HA IP. The immunoprecipitate was then western blotted with anti-HA, anti-Eomes and anti-Tbet antibodies.

Reviewer #3

Zhang and colleagues study the unique or redundant roles for the transcription factors Tbet and Eomes in mouse NK cell differentiation. Previous genetic ablation studies have demonstrated that both Tbet and Eomes are required for normal NK cell homeostasis and differentiation into functionally mature NK cells. Nevertheless, how these factors work independently or in synergy to promote NK cell development is not fully understood. The authors use a series of techniques to better characterize NK cells in mice lacking Tbet or lacking Eomes in NKp46⁺ cells. These include in

depth FACS-based approaches as well as genome wide transcriptomic and epigenetic studies. Together, this results suggest a complex transcriptional regulation mediated by these two related factors that includes both redundant as well as unique roles.

The data presented in Figure 1A suggests that gene dosage affects Tbet and Eomes protein expression differentially. For example, Tbet heterozygous mice have half the expression levels of Tbet protein whereas this is not the case for Eomes heterozygotes. The authors should provide statistical analysis comparing protein levels in WT versus heterozygous mice and discuss this point more fully.

We had indeed missed this point in the previous version of the manuscript. We added a statistical analysis in Figure 1A, and we mentioned this observation in the results section.

The Imagestream analysis (Fig 1D) shows three representative cells. The authors should provide quantification and statistical analysis of a larger number of cells to support this point.

We think the reviewer confused the panels since Figure 1C showed the representative Image-stream analysis, and Figure 1D the statistical analysis for this experiment.

The data in Fig 1D shows inverse relationship of nuclear Tbet and Eomes in different NK cell subsets (derived from confocal studies); it is not clear how many cells are analyzed for each subset (one cell, many cells with average MFI???) and why subsets are connected with lines. This requires explanation.

We agree that this panel was misleading. We changed the graph to make it clearer that many cells were analyzed. We also mentioned the number of cells analyzed in the figure legend (between 1232 and 22403, depending on the subset).

The co-localization of Tbet and Eomes (Fig 1E) requires correlation analysis that should be provided.

We now provided a correlation analysis (Figure S1D) showing overall low correlation between all three parameters analyzed (DAPI, T-bet and Eomes). Yet, the best correlations were observed between T-bet and Eomes, both in immature and mature NK cells. We modified the text accordingly.

The results in Fig 2A-C are largely consistent with previous published and more recent reports (ref 32) that Eomes plays a key role in CD11b⁻ to DP stage (and beyond). The adoptive transfer studies (Fig 2D) however show that Eomes KO CD11b⁻ cells progress to DP and CD27⁻ stages. The authors interpret this as 'accelerated maturation' but this seems at odds with the data in Fig 2A-C as CD27⁻ NK cells are not present in the absence of Eomes. The authors analyze mice two weeks after adoptive transfer; how many cells are recovered? Are these cells Eomes deficient?

We agree with the reviewer that there is an apparent discrepancy between these observations; ie there are very few NK cells in the absence of Eomes but the remaining mature cells have an "hypermature" phenotype (ie increased expression of KLRG1 and other maturation markers). We postulate that even though most NK cells can not develop in the absence of Eomes and therefore die by apoptosis (as shown in Figure 3C), a small fraction of them can survive and become hyper mature

because of the excess of T-bet in the absence of Eomes. Following the reviewer's recommendation, we calculated the recovery rate in the transfer experiment. Indeed, this rate was higher for the WT NK cells, confirming the poor viability of Eomes KO NK cells. The recovered cells were however still Eomes KO. We added these important informations in Figure S2C-D.

Data in Fig 3 suggest Tbet and Eomes suppress proliferation and control IL-15 responses. Are any particular NK cell subsets more or less susceptible with respect to survival in different IL-15 concentrations? It would be useful to know the phenotype of NK cells that survive in the presence of IL-15 in WT versus Tbet KO versus Eomes KO.

To address this point, we analyzed CD11b/CD27 expression together with viability after stimulation with IL-15 in the conditions of figure 3C (48h of culture). Results are now presented in figure S3C and show that CD27- NK cells are the less likely to survive, regardless of their genotype. However, deficiency in T-bet further compromised their survival in our culture system. Moreover, Eomes-deficient DP NK cells were also less viable than control DP NK cells in the IL-15 culture. The latter further confirms previous results that Eomes was important for DP NK cell survival⁴. Appropriate changes were made in the text.

The datasets in Fig 4-7 provide a rich resource for how Tbet and Eomes regulate NK cell signatures and functions. Several targets are confirmed at protein or functional levels. It is somewhat surprising that Eomes repression of cell cycle or proliferation was not revealed by this analysis, considering the data in Fig 3. Further discussion of this point is warranted.

We thank the reviewer for this suggestion. As the "metascape" functional annotation did not retrieve terms associated with proliferation, we performed a "gene set enrichment analysis" (GSEA)⁵ to specifically look for an enrichment of cell proliferation. As expected, this term was enriched in Eomes deficient CD27- NK cells (but not in CD11b- NK cells), as shown for the reviewer below (Figure R2). We did not include these data in the manuscript because our adoptive transfer analysis (requested by Reviewer #1, see response to point 1) clearly showed that the increased proliferation of NK cells in the absence of Eomes was cell extrinsic, probably due to an increased amount of IL-15 available in Eomes deficient mice.

Figure R2: GSEA analysis of cell proliferation in Eomes deficient vs WT CD27- NK cells using the RNAseq data in our article. This analysis shows an enrichment of this term in CD27- NK cells.

The novel mouse models provided by tagging Tbet and Eomes loci with HA-V5 sequences are quite interesting (Fig 8). As these are the first description of these mice, it would be important to have a better characterization of them with respect to NK cell development. Some analysis of the impact (if any) of the observed changes in Tbet and/or Eomes expression on NK cell phenotypes, homeostasis and function should be included.

We agree with the reviewer that this was an important control experiment. We analyzed NK cell frequency in the spleen and bone marrow, and spleen NK cell maturation in HA-V5-tagged animals. Results are presented in Figure S8C-D and show that the frequency of NK cells is comparable in HAV5-tagged and control mice. NK cell maturation was however significantly altered in Tbet-HAV5 mice with a reduction of mature NK cells. This is presumably due to the reduction in Tbet expression in those mice, but this phenotype is much less severe than that in Tbet KO mice. This drawback of the model has to be put in balance of the advantage offered by the HAV5 tag for the side-by-side comparison of Tbet and Eomes binding.

The number of binding sites was quite large for both factors and the peak profiles look quite similar for the examples shown in the supplemental figures. One wonders about the specificity of the antibodies used. Have the authors performed control ChIP-seq on WT mice? It is interesting that canonical binding motifs were not the same in Eomes and Tbet peaks; did these profiles change if one considers whether the peaks were Eomes-specific, Tbet-specific or shared?

As detailed in the response to Reviewer#1, we changed the parameters of the peak calling analysis to use a more stringent method to identify Tbet and Eomes binding sites. This resulted in a lower number of peaks, but we would like to stress that these numbers are classical for this type of genome-wide analysis, and in any case, largely inferior to the number of sites that would result from a non-specific chromatin IP. We indeed performed a ChIP-seq analysis with WT NK cells, which is now used as control for the peak-calling analysis. With this new analysis, there is now a new set of detected TF motifs under Eomes and Tbet peaks, and the T-box is among them for both datasets. However, there are still some differences (eg Fli1 for Eomes, STAT4 for Tbet), that provide some interesting biological hypotheses. We tried to analyze separately peaks that appear more specific to either Eomes or Tbet but the number of peaks was too low and therefore the MEME analysis did not retrieve significant results.

References

1. Slattery, M. *et al.* Absence of a simple code: how transcription factors read the genome. *Trends Biochem. Sci.* **39**, 381–399 (2014).
2. Shih, H.-Y. *et al.* Developmental Acquisition of Regulomes Underlies Innate Lymphoid Cell Functionality. *Cell* **165**, 1120–1133 (2016).
3. Bentsen, M. *et al.* ATAC-seq footprinting unravels kinetics of transcription factor binding during zygotic genome activation. *Nat. Commun.* **11**, 4267 (2020).
4. Wagner, J. A. *et al.* Stage-Specific Requirement for Eomes in Mature NK Cell Homeostasis and Cytotoxicity. *Cell Rep.* **31**, 107720 (2020).

5. Subramanian, A. *et al.* Gene set enrichment analysis: A knowledge-based approach for interpreting genome-wide expression profiles. *Proc. Natl. Acad. Sci.* **102**, 15545–15550 (2005).

Reviewer #1 (Remarks to the Author):

I congratulate the authors for the excellent revision

Reviewer #2 (Remarks to the Author):

The revised manuscript is substantially improved and addresses each of my original concerns and comments. I have no further concerns. This is an excellent and exciting study that will likely have a significant (positive) impact on the field.

Reviewer #3 (Remarks to the Author):

The authors have adequately addressed my previous critiques as well as those made by the other reviewers. The revised manuscript includes new results that strongly support the authors conclusions.